# Learning Molecular Representation in a Cell

**Gang Liu**[1], **Srijit Seal**[2], **John Arevalo**[2], **Zhenwen Liang**[1]
**Anne E. Carpenter**[2], **Meng Jiang**[1], **Shantanu Singh**[2]
[1]University of Notre Dame  [2]Broad Institute of MIT and Harvard
`{gliu7, zliang6, mjiang2}@nd.edu`
`{seal, jarevalo, anne, shantanu}@broadinstitute.org`

## Abstract

Predicting drug efficacy and safety *in vivo* requires information on biological responses (e.g., cell morphology and gene expression) to small molecule perturbations. However, current molecular representation learning methods do not provide a comprehensive view of cell states under these perturbations and struggle to remove noise, hindering model generalization. We introduce the **Info**rmation **Align**ment (InfoAlign) approach to learn molecular representations through the information bottleneck method in cells. We integrate molecules and cellular response data as nodes into a context graph, connecting them with weighted edges based on chemical, biological, and computational criteria. For each molecule in a training batch, InfoAlign optimizes the encoder's latent representation with a minimality objective to discard redundant structural information. A sufficiency objective decodes the representation to align with different feature spaces from the molecule's neighborhood in the context graph. We demonstrate that the proposed sufficiency objective for alignment is tighter than existing encoder-based contrastive methods. Empirically, we validate representations from InfoAlign in two downstream applications: molecular property prediction against up to 27 baseline methods across four datasets, plus zero-shot molecule-morphology matching. The code and model are available at `https://github.com/liugangcode/InfoAlign`.

## 1 Introduction

Drug properties, e.g., toxicity and adverse effects (Liu et al., 2023a), are induced by molecular initiating events—interactions between a molecule and a biological system—that first impact the cellular level and ultimately influence tissue or organ functions (Mast et al., 2014). However, a chemical molecule's structure alone is insufficient information to predict its impact on cells: each chemical interacts with multiple cells and genes and induces complex changes in gene expression and cell morphology, making predictions of downstream responses challenging (Carpenter et al., 2006; Moshkov et al., 2023). Hence, *molecular representation learning should make use of information about cellular response*, enhancing the representation of the mode of action and thereby improving predictions for downstream bioactivity tasks (Liu et al., 2023a; Wang et al., 2023a).

There is a lack of exploration for holistic molecular representations from molecular structure, cell morphology, and gene expression (Hu et al., 2020a; You et al., 2020; Liu et al., 2022; Wang et al., 2023a; Sanchez-Fernandez et al., 2023). For example, graph self-supervised methods only manipulate molecular structures to perturb or mask molecular graphs using contrastive or predictive losses (Hu et al., 2020a; You et al., 2020; Inae et al., 2023). Moshkov et al. (2023) explored the ability of different data modalities, taken independently, to predict molecules' assay activity in a diverse set of assays (tasks). They found (from (Moshkov et al., 2023)'s Fig.2) that molecular structure supports highly accurate prediction (AUC > 90%) in 31% (16/52) of tasks, gene expression in 37% (19/52) and cell morphology in 54% (28/52). Similarly, in our experiments (Figure 3), we observe that molecular structure is not a one-size-fits-all solution.

Cells can be perturbed by treating them with chemicals or genetic reagents that disrupt a particular gene or pathway. These chemical and genetic perturbations *in vitro* naturally bridge molecules with cell morphology and gene expression, as illustrated in Figure 1 (b). However, multi-modal contrastive methods such as CLOOME (Sanchez-Fernandez et al., 2023) and InfoCORE (Wang et al.,

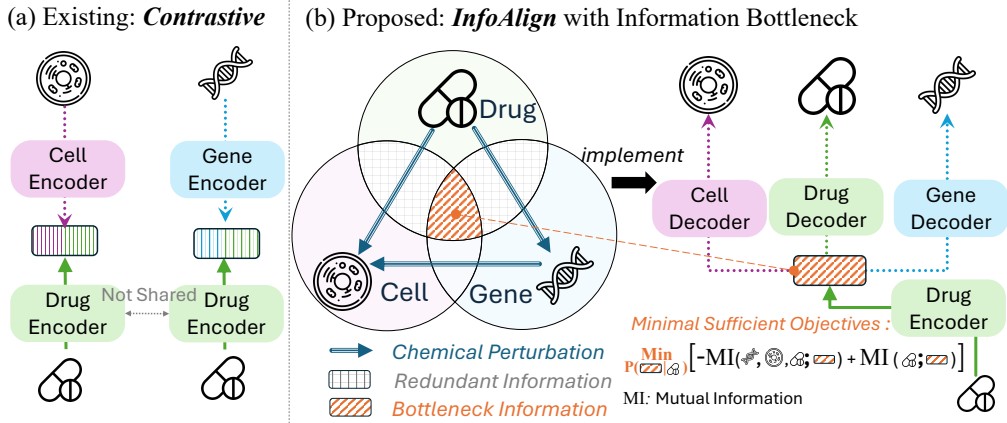

Figure 1: Comparison of Representation Learning Methods: (a) Existing contrastive methods use two encoders—one for molecules and another for cell morphology or gene expression features—without sharing the molecule encoders for different alignment targets. (b) InfoAlign remove redundant information from molecules, cell morphology, and gene expressions based on the information bottleneck, resulting in more concise yet predictive molecular representations (Alemi et al., 2016).

2023a), depicted in Figure 1 (a), focus primarily on aligning molecular representations with cell morphology (Sanchez-Fernandez et al., 2023; Wang et al., 2023a) or gene expression (Wang et al., 2023a). These approaches fall short in two ways. (1) They do not remove redundant information, grey-colored area in Figure 1 (b), that may harm representation generalization. The presence of redundant information (Wang et al., 2023a) may induce spurious correlations, adversely affecting the generalization of molecular representations. For example, in small molecule perturbations (Bray et al., 2016; Chandrasekaran et al., 2023), batch identifiers can signify confounding technical factors, creating misleading associations between molecular structures and cell morphology (Wang et al., 2023a). (2) They treat molecules as the sole connectors between gene expression and cell morphology, ignoring the potential for genetic perturbations (Chandrasekaran et al., 2023) to directly establish connections between these modalities. Genetic perturbations (Chandrasekaran et al., 2023) not only enrich the feature space of gene expression and cell morphology but also enhance the navigation of molecular representation learning towards the overlapped (bottleneck) area in Figure 1 (b).

To address the aforementioned challenges, we conceptualize the cellular response processes as a context graph, capturing a more complete set of interactions among molecules, gene expression, and cell morphology. We identify the neighborhood of the molecule on the context graph and apply the information bottleneck (Tishby et al., 2000) to optimize molecular representations, which aligns them with neighboring biological variables to remove redundant information and improve generalization.

We propose the **Info**rmation **Align**ment (InfoAlign) approach, as presented in Figure 1 (b). InfoAlign uses one encoder and multiple decoders with information bottleneck for minimal sufficient statistics in representation learning. The minimality objective optimizes the encoder to learn the *minimal* informative representation from molecular structures by discarding redundant information. The sufficiency objective ensures the encoder retains *sufficient* information, allowing decoders to reconstruct features for biological variables in neighborhood areas of the context graph. We construct the context graph based on molecule and genetic perturbations (Bray et al., 2017; Chandrasekaran et al., 2023; Subramanian et al., 2017) and introduce more biological (gene-gene interaction (Himmelstein et al., 2017)) and computational (cosine similarity) criteria to increase edge connectivity. We conduct random walks on the context graph, beginning with the molecule in the training batch, to identify its neighborhood. Cumulative edge weights indicate similarity between the molecule and variables along the path. The molecule is encoded, and its latent representation is decoded to align with features identified in the random walk. Encoders and decoders are jointly optimized using an upper bound for the minimality objective and a lower bound for the sufficiency objective.

The sufficiency objective introduces a decoder-based bound for multi-modal alignment. We show its theoretical advantages by demonstrating that it provides a tighter bound than the encoder-based approaches used in previous contrastive methods (Oord et al., 2018; Radford et al., 2021), as discussed

in Section 4.3. In experiments, InfoAlign outperforms up to **27** baselines across three classification and one regression dataset, covering 685 tasks, with average improvements of up to 6.4%. InfoAlign also demonstrates strong zero-shot multi-modal matching on two molecule-morphology datasets.

## 2 RELATED WORK

**Representation Learning on Molecular Structure:** Representation learning approaches for molecules can be categorized into sequential-based (Krenn et al., 2022; Ross et al., 2022) or graph-based models (Hu et al., 2020a; You et al., 2020; Zhang et al., 2021; Liu et al., 2023b). Sequential models, utilizing string formats of molecules like SMILES and SELFIES (Krenn et al., 2022), have evolved from Recurrent Neural Networks (RNNs) to Transformers (Chithrananda et al., 2020; Ross et al., 2022). These models typically follow specific pretraining strategies similar to language models such as BERT (Devlin et al., 2018), RoBERTa (Liu et al., 2019; Chithrananda et al., 2020) and GPT (Radford et al., 2019). The pretraining targets are thus often the next token predictions or mask language modeling (Devlin et al., 2018; Chithrananda et al., 2020) on SMILES or SELFIES sequences (Radford et al., 2019). Graph Neural Networks (GNNs) are the architectures for graph-based approaches (Hu et al., 2020a; You et al., 2020; Zhang et al., 2021; Liu et al., 2024b), where methods to pretrain GNNs often perturb or mask the atoms, edges, or substructures of molecular graphs with contrastive (Hu et al., 2020a; You et al., 2020) and predictive losses (Zhang et al., 2021; Inae et al., 2023). Recent evidence highlights the challenges of developing universal molecular representations based solely on molecular structures without integrating domain knowledge (Bray et al., 2016; Seal et al., 2022; Sun et al., 2022; Seal et al., 2023; Liu et al., 2024a). Although using motifs is a common method to incorporate such knowledge (Rong et al., 2020; Inae et al., 2023), the incorporation of information about molecules' biological impacts is much less explored. We aim to enhance molecular representation learning by incorporating domain knowledge from cellular response data.

**Representation Learning with Cellular Response Data:** A primary goal of molecular representation learning is to predict molecular bioactivity. Likewise, emerging gene expression Subramanian et al. (2017) and morphological profiling approaches Carpenter et al. (2006); Seal et al. (2024) that describe perturbed genetic or cellular states in cell cultures can also be used to predict bioactivity. In some datasets, molecules are the perturbations, and the perturbed cell states measured are gene expression values for a thousand or more genes (Subramanian et al., 2017) and/or microscopy Cell Painting images, which can be represented as a thousand or more morphology features Cimini et al. (2023). Recently created large-scale perturbation datasets (Subramanian et al., 2017; Chandrasekaran et al., 2023) could enrich molecular representation learning approaches. CLOOME (Sanchez-Fernandez et al., 2023), MIGA (Zheng et al., 2024), MoCoP (Nguyen et al., 2023), and MolPhenix (Fradkin et al., 2024) contrast cellular images with molecules. InfoCORE (Wang et al., 2023a) contrasts molecule with either morphological profiling (Bray et al., 2017) or gene expression (Wang et al., 2023a). InfoCORE Wang et al. (2023a) mitigates confounding batch identifiers using a batch classifier, which may not be practical when batch identifiers are unavailable during training.

## 3 PROBLEM DEFINITION

We denote $x \in \mathcal{X}$ as the molecule from the space $\mathcal{X}$. An encoder with parameters $p_\theta(\mathbf{z} \mid x)$ maps $x$ to a $D$-dimensional latent representation $\mathbf{z} \in \mathbb{R}^D$. One may implement a Graph Neural Network (GNN) (Xu et al., 2019) as the encoder. The GNN first updates node representations and then performs a readout operation (e.g., summation) over the nodes to obtain the latent representation.

Existing research has extensively used structural features to pretrain the GNN encoder (Hu et al., 2020a; Inae et al., 2023). However, incorporating more expressive features from the cellular context, such as cell morphology and gene expression, remains largely unexplored for improving molecular representations. In this work, we use these features as targets to optimize molecular representations.

## 4 MULTI-MODAL ALIGNMENT WITH INFOALIGN

We present the overall representation learning framework in Figure 2. In Section 4.1, we construct the context graph for cellular response data. In Section 4.2, we introduce representation learning

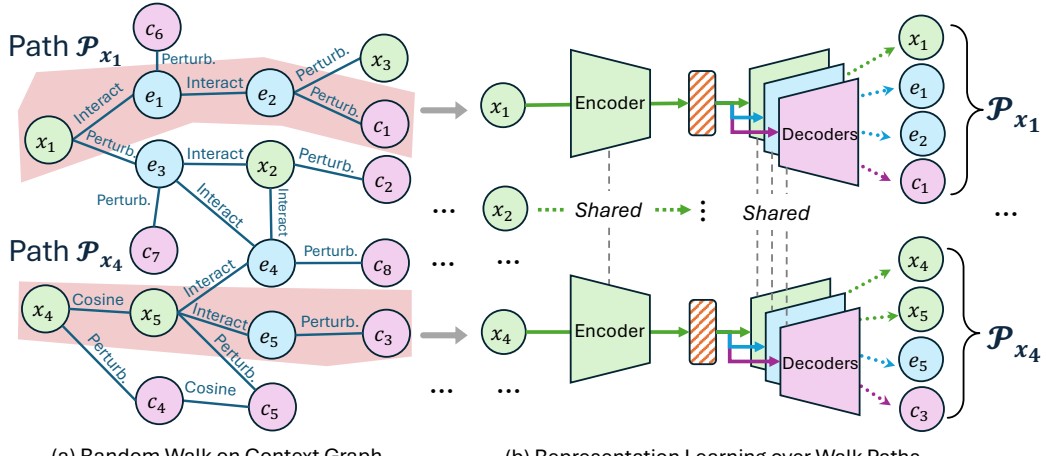

(a) Random Walk on Context Graph

(b) Representation Learning over Walk Paths

Figure 2: Molecular Representation Learning Using the Context Graph: (a) In Section 4.1, we construct the graph with various interaction, perturbation, and cosine similarities among molecules $x$, cell morphology profiles $c$, and genes $e$. Given a training batch of molecules, including $x_1$ and $x_4$, random walk extracts paths, for instance, of length four. (b) In Section 4.2, we aim to learn molecular representations based on the information bottleneck, preserving minimal information from the input molecule while ensuring sufficient information for decoding the target along the walk path $\mathcal{P}_x$.

methods based on the principle of minimal sufficiency for molecules and their related modalities. In Section 4.3, we demonstrate the theoretical advantages of the proposed method.

## 4.1 RANDOM WALKS ON CELLULAR CONTEXT GRAPH

**Node Construction:** We model the interactions of the molecule $x$ with other molecules, the cell $c$, and genes $e$ using the context graph. They are nodes with different features $y$. Node features for molecules are binary vectors obtained using fingerprints (Rogers & Hahn, 2010). Cell morphology features are derived from CellProfiler (Carpenter et al., 2006) applied to Cell Painting microscopy images. Gene nodes have expression values obtained using L1000 methods (Subramanian et al., 2017). We rescale the cell morphology and gene expression features to a range between 0 and 1.

**Edge Construction:** We link nodes using chemical, biological, and computational criteria. For example, molecules can perturb cultured human cells, inducing changes in cell morphology (Chandrasekaran et al., 2023) and gene expression (Subramanian et al., 2017), thus linking molecules to cell morphology and gene expression nodes. Genes could also perturb cells, inducing links between genes and cell morphology (Chandrasekaran et al., 2023). Additionally, We calculate cosine similarity for nodes of the same type and use biological criteria, such as gene-gene interactions (Himmelstein et al., 2017), to enrich edges. Each edge is assigned a weight $w$ ranging from 0 to 1. For example, edges derived from computational criteria between molecule nodes are assigned weights based on the assumption that structurally similar molecules may exhibit similar biological effects, a concept widely used in drug discovery, such as lead optimization. We construct the context graph as detailed in Section 5 and appendix C.1, with an example shown in Figure 2 (a).

**Random Walk Path Extraction:** The context graph identifies related cellular response patterns for input molecules in representation learning. Given an input molecule $x$, we extract its neighborhood through random walks starting from $x$. Specifically, we employ degree-based transition probabilities (Perozzi et al., 2014) and denote the walking path as $\mathcal{P}_x : x \xrightarrow{w_1} v_2 \xrightarrow{w_2} \ldots \xrightarrow{w_L} v_L$, where $v_2$ is a direct neighbor of $x$. To quantify the similarity between $x$ and node $v_i$ ($2 \leq i \leq L$) on $\mathcal{P}_x$, we compute the cumulative product of edge weights as $\alpha(v_i \mid \mathcal{P}_x) = \prod_{j=1}^{i-1} w_j$.

## 4.2 OPTIMIZATION FOR REPRESENTATION WITH INFORMATION BOTTLENECK

The information bottleneck (IB) (Tishby et al., 2000; Alemi et al., 2016) is an appealing method for defining concise representations with strong predictive power. For molecular representation, we

extract minimal sufficient information from the random variable $X$ of molecules. This is achieved by aligning the molecular representations $Z$ with the targets $Y$, derived from node features along the walk path $\mathcal{P}$. The IB has two principles based on mutual information (MI): (1) the minimality principle, which minimizes MI between molecules and their latent representations as $I(X; Z)$, and (2) the sufficiency principle, which decodes latent representations to maximally reconstruct feature spaces for variables along the walk path $I(Z; Y)$. Together, these form the optimization objectives:

$$\min_{p(\mathbf{z}|x)} \left[ -I(Z; Y) + \beta I(X; Z) \right], \tag{1}$$

where $\beta$ controls the trade-off between minimality and sufficiency. The exact computation of $I(Z; Y)$ and $I(X; Z)$ is intractable due to the unknown conditional distribution $p(y|\mathbf{z})$ and the marginal $p(\mathbf{z})$. We introduce the variational approximations $q(y|\mathbf{z})$ and $q(\mathbf{z})$ for them, respectively. This results in a lower bound estimation for the first decoding term $I_{DLB}$ and an upper bound for the second encoding term $I_{EUB}$ (Poole et al., 2019).

$$
\begin{aligned}
I(Z; Y) &\geq \mathbb{E}_{p(\mathbf{z},y)} \left[ \log q(y \mid \mathbf{z}) \right] + H(Y) \triangleq I_{DLB} \\
I(X; Z) &\leq \mathbb{E}_{p(x)} \left[ \mathrm{KL} \left( p(\mathbf{z} \mid x) \| q(\mathbf{z}) \right) \right] \triangleq I_{EUB}
\end{aligned}
\tag{2}
$$

$H(Y)$ is the differential entropy. Proofs are in appendix B.1. Together, $I_{DLB}$ and $I_{EUB}$ upper bound Eq. (1), forming a tractable objective $-I_{DLB} + I_{EUB}$ to optimize the encoder. For the target $Y$, the $I_{DLB}$ objective requires decoders rather than encoders, as typically used in prior work (Sanchez-Fernandez et al., 2023). We use distinct decoders, denoted as $q_\phi$ with parameters $\phi$, for various targets, including molecular fingerprints, gene expressions, and cell morphology features.

After ignoring the constant terms, one could formulate the loss function according to Eq. (2) for the molecule sample $x$, its latent representation $\mathbf{z}$, and the targets $y_v$ from $\mathcal{P}_x$:

$$\mathcal{L} = \frac{1}{L} \sum_{v \in \mathcal{P}_x} \alpha(v|\mathcal{P}_x) \left[ -\log \left( q_\phi(y_v \mid \mathbf{z}) \right) + \beta \, \mathrm{KL} \left( p_\theta(\mathbf{z} \mid x) \mid \mathcal{N}(0, I) \right), \tag{3}$$

where the first term aligns the representation with other features, and KL is the Kullback–Leibler divergence used for regularization. $\mathcal{N}(0, I)$ is the Gaussian prior. In this formulation, the encoder models a distribution instead of a single representation $\mathbf{z}$, learning the mean and variance $\boldsymbol{\mu}, \boldsymbol{\sigma} \in \mathbb{R}^D$. One may use parameterization tricks to sample $\mathbf{z}$ from the distribution (Alemi et al., 2016). The decoder then reconstructs $y_v$, the features of the neighboring node $v \in \mathcal{P}_x$ from the walk path.

InfoAlign uses multiple decoders for $q_\phi$ to align multi-modal features, while prior work relies on encoders with CLIP-like losses to align the latent space (Radford et al., 2021; Girdhar et al., 2023; Wang et al., 2023a; Sanchez-Fernandez et al., 2023). Next, we provide the theoretical benefits of decoder-based alignment alongside the empirical advantages in Section 6.

### 4.3 Theoretical Motivation for Decoder-based Alignment

InfoNCE (Oord et al., 2018) is the contrastive loss used for most CLIP-like methods (Radford et al., 2021; Wang et al., 2023a). In this work, we show that the MI lower bound based on InfoAlign is tighter than that based on InfoNCE.

**Proposition 4.1.** *For the molecular representation $Z$ and target $Y$ (from cell morphology, gene expressions, or molecular fingerprints), the encoder-based MI lower bound $I_{ELB}$ for InfoNCE can be derived by incorporating $K - 1$ additional samples, denoted as $y_{2:K}$, to build the Monte Carlo estimate $m(\cdot)$ of the partition function (Nguyen et al., 2010; Poole et al., 2019):*

$$I_{ELB} = 1 + \mathbb{E}_{p(z,y)p(y_{2:K})} \left[ \log \frac{e^{h(z,y)}}{m(z; y, y_{2:K})} \right] - \mathbb{E}_{p(z)p(y_{2:K})p(y)} \left[ \frac{e^{h(z,y)}}{m(z; y, y_{2:K})} \right], \tag{4}$$

*where $h(z, y)$ is the neural network parameterized critic for density approximation with the energy-based variational family. The decoder-based lower bound $I_{DLB}$ is defined in Eq. (2), then we have that $I_{DLB}$ is tighter than $I_{ELB}$, i.e., $I(Z; Y) \geq I_{DLB}(Z; Y) \geq I_{ELB}(Z; Y)$.*

Proofs are in appendix B.2. The result aligns with empirical observations in previous studies such as DALL-E 2 (Ramesh et al., 2022), where a prior model was introduced to improve representations from CLIP (Radford et al., 2021) before decoding to another modality. In this work, we learn decodable latent representations from molecules to align with different biological features.

## 5    IMPLEMENTATION OF CONTEXT GRAPH AND PRETRAINING SETTING

**Data Source of Context Graph:** We create the context graph based on (1) two Cell Painting datasets (Bray et al., 2017; Chandrasekaran et al., 2023), containing around 140K molecule perturbations (molecule and cell morphology pairs) and 15K genetic perturbations (gene and cell morphology pairs) across 1.6 billion human cells; (2) Hetionet (Himmelstein et al., 2017), which captures gene-gene and gene-molecule relationships from millions of biomedical studies; and (3) a dataset reporting differential gene expression values for 978 landmark genes (Wang et al., 2016) for chemical perturbations (molecule and gene expression pairs) (Subramanian et al., 2017).

**Node Features:** Different profiling methods provide node features in different ways. Morgan fingerprints (Rogers & Hahn, 2010) are feature vectors extracted from each molecule's structure, CellProfiler features (Carpenter et al., 2006) are computed from the image of each cell and represent cell morphology, and L1000 profiles (Subramanian et al., 2017) capture gene expression values on 978 landmark genes from cells treated with a chemical perturbation. Here are two practical considerations for the context graphs: (1) Chandrasekaran et al. (2023) provided one dataset that measured the cell morphology impacts of perturbing individual genes. The 15K genetic perturbations (Chandrasekaran et al., 2023) provide gene-cell morphology pairs but lack corresponding gene expression profiles. Still, we keep the gene nodes from this dataset to account for potential gene-gene interactions and incorporate cell morphology features into them. (2) All 978 landmark genes have expression values linked to the molecules used in (Wang et al., 2016). We update new gene expression nodes with 978-dimensional feature vectors. These vectors summarize all molecule-gene expression connections for a small molecule perturbation. This approach efficiently reduces dense connections between landmark genes and molecules. We select the top 1% of gene-molecule expression values as new edges to enrich the context graph's connectivity. We scale cell morphology and gene expression features to a range of 0 to 1 using the Min-Max scaler along each dimension.

**Edge Weights:** For edges based on chemical perturbations (Chandrasekaran et al., 2023), we assign the edge weight of 1. We also compute cosine similarity for nodes if they are in the same feature space (such as two cell morphology/gene expression profiles, or Morgan fingerprints). To avoid noisy edges from computations, we (1) apply a 0.8 threshold for cosine similarity, and additionally (2) explicitly enforce 99.5% sparsity by selecting top similar edges.

All together, this results in a context graph of 276,855 nodes (129,592 molecules, 4533 genes + 13,795 gene expressions, and 128,935 cell morphology) and 366,384 edges.

**Encoder and Decoder:** We use a five-layer Graph Isomorphism Network (GIN) (Xu et al., 2019) with sum as the readout function as the molecule encoder. All molecules on the context graph are used to pretrain the encoder. Since we extract feature vectors as decoding targets in different modalities, we efficiently use a Multi-Layer Perceptron (MLP) as modality decoders. We set the hidden dimension to 300, $\beta = 4$, and $L = 4$. In each training batch, random walks start from the molecule node to extract the walk path. The decoders are then pretrained to reconstruct the corresponding node features along the path. Further details can be found in Section appendix C.

## 6    EXPERIMENTS

We demonstrate the effectiveness of InfoAlign's representation in (1) molecular property prediction, (2) molecule-morphology matching, and (3) analyze the performance of InfoAlign. These lead to three research questions (RQs).

### 6.1    RQ1: MOLECULAR PROPERTY PREDICTION

Better molecular representations should improve prediction performance. We train MLPs on different representations to predict molecular properties in both classification and regression tasks.

### 6.1.1    EXPERIMENTAL SETTING

**Dataset and Evaluation:** We select datasets for important tasks in drug discovery, including activity classification for various assays in ChEMBL2K (Gaulton et al., 2012) and Broad6K (Moshkov et al., 2023), drug toxicity classification using ToxCast (Richard et al., 2016), and absorption, distribution, metabolism, and excretion (ADME) regression using Biogen3K (Fang et al., 2023). The dataset

Table 1: Results on ChEMBL2K and Broad6K. We report average AUC (Avg.), as well as the percentage of tasks achieving AUC above 80%, 85%, and 90%. We highlight the **best** and second best mean. We also highlight the  row of the best method  in each category.

| Dataset (# Molecule / # Task) Method | ChEMBL2k (AUC ↑) (2355 / 41) | | | | Broad6k (AUC ↑) (6567 / 32) | | | |
|---|---|---|---|---|---|---|---|---|
| | Avg. | >80% | >85% | >90% | Avg. | >80% | >85% | >90% |
| *Morgan Fingerprints* | | | | | | | | |
| MLP | 76.8±2.2 | 48.8±3.9 | 34.6±6.3 | 21.9±5.7 | 63.3±0.3 | 6.3±0.0 | **4.4**±1.7 | **3.1**±0.0 |
| RF | 54.7±0.7 | 0.0±0.0 | 0.0±0.0 | 0.0±0.0 | 55.5±0.1 | 0.0±0.0 | 0.0±0.0 | 0.0±0.0 |
| GP | 51.0±0.0 | 0.0±0.0 | 0.0±0.0 | 0.0±0.0 | 50.6±0.0 | 0.0±0.0 | 0.0±0.0 | 0.0±0.0 |
| *Pretrained GNN* | | | | | | | | |
| AttrMask (Hu et al., 2020a) | 73.9±0.5 | 46.8±2.7 | 31.2±4.4 | 14.6±1.7 | 59.8±0.2 | 3.1±0.0 | 3.1±0.0 | **3.1**±0.0 |
| ContextPred (Hu et al., 2020a) | 77.0±0.5 | 55.1±1.3 | 34.1±4.6 | 14.6±1.7 | 60.0±0.2 | 7.5±1.7 | 3.1±0.0 | **3.1**±0.0 |
| EdgePred (Hu et al., 2020a) | 75.6±0.5 | 54.2±4.0 | 34.6±7.2 | 12.2±2.4 | 59.9±0.2 | 3.1±0.0 | 3.1±0.0 | **3.1**±0.0 |
| GraphCL (You et al., 2020) | 75.6±1.6 | 46.8±7.6 | 32.2±6.8 | 18.0±3.7 | 67.2±0.5 | 15.6±3.1 | 3.1±0.0 | **3.1**±0.0 |
| GROVER (Rong et al., 2020) | 73.3±1.4 | 38.5±2.0 | 22.4±3.6 | 14.6±2.4 | 66.2±0.1 | 15.6±0.0 | 3.8±1.4 | **3.1**±0.0 |
| JOAO (You et al., 2020) | 75.1±1.0 | 47.8±5.1 | 33.7±2.0 | 19.0±3.2 | 67.3±0.4 | 12.5±0.0 | 3.8±1.4 | **3.1**±0.0 |
| MGSSL (Zhang et al., 2021) | 75.1±1.1 | 39.0±4.6 | 29.3±3.0 | 10.3±3.2 | 66.9±0.5 | 13.8±2.8 | 3.1±0.0 | **3.1**±0.0 |
| GraphLoG Xu et al. | 73.5±0.7 | 41.9±2.0 | 29.3±3.4 | 15.6±2.8 | 62.9±0.4 | 4.4±1.7 | 0.0±0.0 | 0.0±0.0 |
| GraphMAE (Hou et al., 2022) | 74.7±0.1 | 33.2±1.3 | 27.8±1.3 | 12.2±1.7 | 66.8±0.3 | 14.4±1.7 | 3.1±0.0 | **3.1**±0.0 |
| DSLA (Kim et al., 2022) | 69.3±1.0 | 23.9±4.7 | 14.6±5.5 | 6.8±1.1 | 63.3±0.3 | 6.3±0.0 | 3.1±0.0 | **3.1**±0.0 |
| UniMol (Zhou et al., 2023) | 76.8±0.4 | 46.8±2.0 | 33.7±1.1 | 24.9±2.0 | 65.4±0.1 | 7.5±1.7 | 3.1±0.0 | **3.1**±0.0 |
| *Pretrained Chemical Language Models* | | | | | | | | |
| Roberta (Mary et al., 2024) | 74.7±1.9 | 46.3±3.4 | 35.1±4.4 | 22.9±1.3 | 59.8±0.7 | 5.0±1.7 | 3.1±0.0 | **3.1**±0.0 |
| GPT2 (Mary et al., 2024) | 71.0±3.4 | 31.2±11.2 | 20.0±9.4 | 7.3±6.9 | 60.6±0.3 | 7.5±1.7 | 1.9±1.7 | 1.9±1.7 |
| MolT5 (Edwards et al., 2022) | 69.9±0.8 | 32.2±2.0 | 21.0±4.1 | 8.8±1.3 | 56.4±0.8 | 3.8±1.4 | 2.5±1.4 | 2.5±1.4 |
| ChemGPT (Frey et al., 2023) | 65.0±1.1 | 16.1±2.8 | 11.2±3.3 | 5.4±1.1 | 55.1±0.9 | 3.1±0.0 | 3.1±0.0 | 1.3±1.7 |
| *Cell Morphology* | | | | | | | | |
| MLP | 64.3±2.4 | 15.6±6.6 | 8.3±3.7 | 4.9±3.9 | 51.9±1.0 | 0.0±0.0 | 0.0±0.0 | 0.0±0.0 |
| RF | 55.9±0.7 | 3.9±1.3 | 3.9±1.3 | 2.4±0.0 | 55.3±0.1 | 0.0±0.0 | 0.0±0.0 | 0.0±0.0 |
| GP | 50.1±0.0 | 0.0±0.0 | 0.0±0.0 | 0.0±0.0 | 54.7±0.0 | 0.0±0.0 | 0.0±0.0 | 0.0±0.0 |
| *Gene Expression* | | | | | | | | |
| MLP | 56.1±1.1 | 5.1±1.4 | 3.4±1.3 | 3.4±1.3 | 56.9±1.4 | 1.9±1.7 | 1.9±1.7 | 1.9±1.7 |
| RF | 52.8±0.3 | 0.0±0.0 | 0.0±0.0 | 0.0±0.0 | 55.2±0.2 | 0.0±0.0 | 0.0±0.0 | 0.0±0.0 |
| GP | | Run out of time | | | 50.1±0.0 | 0.0±0.0 | 0.0±0.0 | 0.0±0.0 |
| *Multi-modal Alignment* | | | | | | | | |
| CLOOME | 66.7±1.8 | 26.8±4.6 | 16.1±3.7 | 10.7±5.1 | 61.7±0.4 | 3.1±0.0 | 3.1±0.0 | 0.0±0.0 |
| InfoCore (GE) | 79.3±0.9 | 62.4±2.8 | 46.3±3.0 | 30.3±2.2 | 60.2±0.2 | 3.1±0.0 | 0.0±0.0 | 0.0±0.0 |
| InfoCore (CP) | 73.8±2.0 | 37.6±9.2 | 26.3±4.7 | 10.7±4.1 | 61.1±0.2 | 6.3±0.0 | 3.1±0.0 | 0.0±0.0 |
| InfoAlign (Ours) | **81.3**±0.6 | **66.3**±2.7 | **49.3**±2.7 | **35.1**±3.7 | **70.0**±0.1 | **18.8**±2.2 | 3.1±0.0 | **3.1**±0.0 |

covers **685** tasks with details in Table 5 and appendix D.1. We apply scaffold-splitting for all datasets. We follow a 0.6:0.15:0.25 ratio for training, validation, and test sets for all datasets. We use the Area under the curve (AUC) for classification and mean absolute error (MAE) for regression. Mean and standard deviations are reported from ten runs.

**Baseline:** We include **27** baselines across six categories: (1) three molecular fingerprint (FP)-based methods (Rogers & Hahn, 2010); (2) eleven pretrained GNNs; (3) four pretrained chemical language models; (4,5) six methods based on cell morphology and gene expression values from cells treated with each molecule; (6) CLOOME (Sanchez-Fernandez et al., 2023) and InfoCORE (Wang et al., 2023a) for multi-modal alignment using structure, morphology, and gene expression data. We use MLPs, Random Forests (RF), and Gaussian Processes (GP) for methods in categories (1,4,5). Setting details and all results are in appendices D.1 and D.3.

### 6.1.2 RESULTS AND ANALYSIS

We present results across various assays in Tables 1 and 2 and Figure 3. Key observations include:

(1) **Molecular structures are superior compared to cell morphology and gene expression features for predicting various molecular assays**. This is likely because the datasets and tasks we selected

Table 2: Results on ToxCast and Biogen3K. We report the average AUC and the percentage of AUC above 80% on ToxCast, and regression MAE (scaled by × 100) for Biogen3K. We highlight the **best** and second best mean. We also highlight the  row of the best method  in each category.

| Dataset (# Molecule / # Task) Method | ToxCast (AUC ↑) (8576 / 617) | | Biogen3K (MAE ×100 ↓) (3521 / 6) | | | | | | |
|---|---|---|---|---|---|---|---|---|---|
| | Avg. | >80 % | Avg. | hPPB | rPPB | RLM | HLM | ER | Solubility |
| *Morgan Fingerprints* | | | | | | | | | |
| MLP | 57.6±1.0 | 1.6±0.3 | 66.2±2.4 | 66.1±2.6 | 56.8±2.3 | 56.5±4.2 | 74.6±6.2 | 73.7±7.3 | 69.5±3.0 |
| RF | 52.3±0.1 | 0.2±0.1 | 52.8±0.2 | 44.2±0.1 | 44.2±0.1 | 42.0±0.2 | 67.7±0.7 | 66.9±0.9 | 51.6±0.1 |
| GP | Run out of Time | | 60.0±0.0 | 51.3±0.0 | 59.5±0.0 | 49.7±0.0 | 68.8±0.0 | 69.3±0.0 | 61.6±0.0 |
| *Pretrained GNN* | | | | | | | | | |
| AttrMask (Hu et al., 2020a) | 63.1±0.8 | 3.2±1.2 | 67.3±0.3 | 82.4±1.1 | 49.8±0.7 | 51.7±1.0 | **57.9±0.6** | 62.6±0.5 | 99.1±1.2 |
| ContextPred (Hu et al., 2020a) | 63.0±0.6 | 3.3±1.3 | 68.5±7.9 | 44.5±0.4 | 49.7±0.4 | 55.1±2.7 | 61.4±1.8 | 63.1±0.5 | 96.5±3.7 |
| EdgePred (Hu et al., 2020a) | 63.5±1.1 | 4.8±3.0 | 67.8±0.9 | 81.2±10.2 | 48.0±0.5 | 53.5±2.8 | 62.2±1.8 | 62.9±0.7 | 99.1±6.9 |
| GraphCL (You et al., 2020) | 52.2±0.2 | 0.5±0.3 | 53.9±0.6 | 43.8±0.3 | 45.4±0.6 | 40.6±0.5 | 76.7±1.0 | 67.1±2.2 | 49.6±0.3 |
| GROVER (Rong et al., 2020) | 53.1±0.4 | 0.5±0.1 | 54.9±1.6 | 44.5±0.4 | 46.5±0.7 | 41.7±0.6 | 73.2±5.7 | 71.0±4.3 | 52.6±0.3 |
| JOAO (You et al., 2021) | 52.3±0.2 | 0.4±0.1 | 55.0±0.8 | 44.5±0.5 | 47.6±0.5 | 40.6±0.2 | 74.3±2.8 | 71.5±2.6 | 51.4±0.6 |
| MGSSL (Zhang et al., 2021) | 64.2±0.2 | 4.0±0.4 | 53.2±0.3 | 44.8±0.6 | 49.7±0.3 | 41.5±0.2 | 65.6±1.8 | 64.6±0.5 | 52.7±0.5 |
| GraphLoG (Xu et al., 2021) | 58.6±0.4 | 2.5±0.3 | 56.9±0.4 | 49.3±0.3 | 54.8±0.5 | 42.6±0.3 | 66.8±1.7 | 69.0±1.3 | 58.8±0.5 |
| GraphMAE (Hou et al., 2022) | 53.3±0.1 | 0.6±0.1 | 52.8±0.8 | 43.3±0.9 | 51.2±0.8 | 40.9±0.3 | 64.4±2.7 | 65.9±3.8 | 50.9±1.4 |
| DSLA (Kim et al., 2022) | 57.8±0.5 | 0.7±0.1 | 57.9±0.7 | 50.4±0.7 | 53.6±1.1 | 43.3±0.9 | 68.6±1.2 | 70.8±2.0 | 60.9±0.6 |
| UniMol (Zhou et al., 2023) | 64.6±0.2 | 4.8±1.0 | 55.8±2.8 | 50.1±5.2 | 49.9±5.6 | 43.6±1.1 | 65.4±4.9 | 65.8±1.2 | 59.9±6.6 |
| *Pretrained Chemical Language Models* | | | | | | | | | |
| Roberta (Mary et al., 2024) | 64.2±0.8 | 3.1±1.8 | 69.0±2.6 | 71.4±14.5 | 65.1±19.2 | 63.7±24.6 | 67.5±5.2 | 69.9±4.9 | 76.7±13.2 |
| GPT2 (Mary et al., 2024) | 61.5±1.1 | 2.4±0.6 | 74.0±8.5 | 65.4±12.9 | 73.1±20.8 | 54.1±12.9 | 83.2±21.5 | 86.1±19.8 | 81.8±25.5 |
| MolT5 (Edwards et al., 2022) | 64.7±0.9 | 3.6±1.1 | 65.1±0.5 | 76.7±2.1 | 55.9±1.1 | 49.2±1.0 | 70.3±0.8 | 73.1±1.0 | 65.3±1.7 |
| ChemGPT (Frey et al., 2023) | Token Error | | 75.7±8.5 | 59.5±7.3 | 88.8±32.3 | 76.1±11.8 | 84.0±20.6 | 77.2±8.5 | 68.6±7.1 |
| *Multi-modal Alignment* | | | | | | | | | |
| CLOOME | 54.2±0.9 | 0.9±0.2 | 64.3±0.4 | 65.2±1.5 | 56.9±0.8 | 44.2±0.8 | 70.7±0.4 | 73.6±0.8 | 75.0±2.1 |
| InfoCORE (GE) | 65.3±0.2 | 5.4±1.7 | 69.9±1.2 | 79.9±3.6 | 51.6±1.8 | 51.3±2.1 | 78.6±0.3 | 77.8±1.9 | 80.3±0.9 |
| InfoCORE (CP) | 62.4±0.4 | 1.3±0.5 | 71.0±0.6 | 74.5±4.9 | 53.5±0.7 | 53.6±2.1 | 80.8±1.5 | 79.4±3.4 | 84.4±1.0 |
| InfoAlign (Ours) | **66.4±1.1** | **6.6±1.6** | **49.4±0.2** | **39.7±0.4** | **39.2±0.3** | **40.5±0.6** | 66.7±1.7 | **62.0±1.5** | **48.4±0.6** |

(a) ChEMBL2K with 41 Tasks.

(b) Broad6K with 32 Tasks.

Figure 3: Percentage of Tasks Where Representations Excel: The top bar compares the best baselines using single-modal representations (Single Rep.) across representation categories. The bottom bar compares three aligned representations (Aligned Rep.): InfoAlign, CLOOME, and InfoCORE.

fundamentally involve predicting the binding affinity of a molecule to a protein Gaulton et al. (2012); furthermore, in these datasets, molecules with activity in a given assay tend to have highly related structures, rather than representing two or more structurally distinct classes of molecules with activity; together this implies that molecular structure alone will tend to yield strong results. When comparing the three popular structure-based representation approaches, no single method outperforms the others across all four datasets. Pretrained GNNs generally perform better than fingerprint-based methods and pretrained chemical language models, thanks to recent advancements. However, continued efforts in universal structural representation are still necessary.

(2) **Cell morphology and gene expression features may complement molecular structures, yielding more generalizable representations**. As shown in Figure 3, cell morphology and gene expression outperform molecular structure in approximately 20% and 10% of tasks on the ChEMBL2K and Broad6K datasets, respectively. This suggests that incorporating cell context into representation learning would be beneficial. That said, existing multi-modal baselines (InfoCORE, CLOOME)

Table 3: Retrieval results on ChEMBL2K and Broad6K: Ranking metrics for top candidates.

| ChEMBL2K | NDCG % (↑) | | HIT % (↑) | | Broad6K | NDCG % (↑) | | HIT % (↑) | |
|---|---|---|---|---|---|---|---|---|---|
| | top-1 | top-10 | top-1 | top-10 | | top-1 | top-10 | top-1 | top-10 |
| CLOOME | 0 | 2.0 | 0 | 6.3 | CLOOME | 0.5 | 0.9 | 0.5 | 1.5 |
| InfoCORE | 0 | 4.5 | 0 | 11.3 | InfoCORE | **1.0** | **2.5** | **1.0** | 4.6 |
| InfoAlign | **1.3** | **5.7** | **1.3** | **12.5** | InfoAlign | 0.5 | 2.3 | 0.5 | **5.1** |

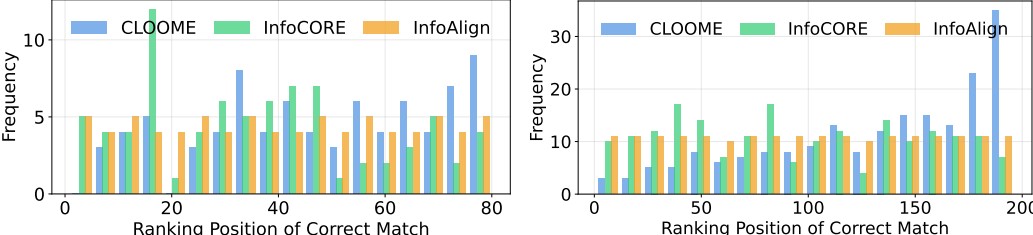

Figure 4: Histogram of rankings for the correct matching on ChEMBL2K (left) and Broad6K (right).

only outperform molecular structure-based approaches on ChEMBL2K and ToxCast, as they do not construct molecular representations holistically by using all cell-related modalities.

(3) **InfoAlign achieves the best average performance on all tasks compared to 27 baselines.** The improvements from InfoAlign range from 2.5% to 6.4% on average across four datasets compared to the second-best method. These gains are more significant when using the 80% AUC threshold on classification datasets. While InfoCORE (GE) performs best among baselines on the ChEMBL2K and ToxCast datasets, it struggles to align molecular representations with more than two modalities and sometimes leads to negative transfer, as seen in Broad6K and Biogen3K.

## 6.2 RQ2: MOLECULE-MORPHOLOGY CROSS-MODAL MATCHING

Molecular representations are aligned with cell morphology. The zero-shot matching performance of a queried molecule to cell morphology features evaluates the alignment between the two modalities.

### 6.2.1 EXPERIMENTAL SETTING

We compared InfoAlign with the CLOOME and InfoCORE (CP) for retrieving cell morphology from molecular representations. We calculate the cosine similarity between the molecular representation and all cell morphology candidates, rank these candidates, and compute Normalized Discounted Cumulative Gain (NDCG) and HIT scores for the top-1 and top-10 candidates as metrics. For a fair evaluation of zero-shot matching, we exclude the cell morphology data for molecules that were used to train the baselines. Consequently, we have 80 molecule-cell morphology pairs from ChEMBL2K and 196 pairs from Broad6K. All the morphology data are used as candidates for matching.

For InfoAlign, we use the pretrained decoder from Section 5 to extract the morphology features of the encoded molecule and then calculate the likelihood of these decoded features against the candidate morphology data. We then rank the candidates in the decoding space based on their likelihood scores.

### 6.2.2 RESULTS AND ANALYSIS

Cross-modal matching results with ranking metrics for the top candidates are in Table 3. InfoAlign outperforms InfoCORE on ChEMBL2K and is comparable on Broad6K, with both surpassing CLOOME. Additionally, we present a histogram of ranking positions for correct matching pairs in Figure 4 to compare overall retrieval performance. The results show that InfoAlign and InfoCORE perform similarly, while CLOOME tends ranks correct pairs lower.

## 6.3 RQ3: PERFORMANCE ANALYSIS

### 6.3.1 ABLATION STUDIES

We perform ablation studies on Eq. (3) by pretraining encoders with different targets removed: (1) molecule-related, (2) cell morphology-related, and (3) gene expression-related features. The results

Table 4: Ablation studies on the pretraining loss. Different node features are removed from the context graph to assess their impact on downstream tasks. Avg. AUC is reported.

|  | ChEMBL2K
AUC ↑ | Broad6K
AUC ↑ | ToxCast
AUC ↑ | Biogen3K
MAE ↓ ($\times 100$) |
|---|---|---|---|---|
| Default as Eq. (3) | $81.3 \pm 0.6$ | $70.0 \pm 0.1$ | $66.4 \pm 1.1$ | $49.4 \pm 0.2$ |
| w/o Cell Morphology | $80.7 \pm 0.6$ | $68.6 \pm 0.1$ | $65.5 \pm 1.1$ | $51.7 \pm 1.1$ |
| w/o Gene Expressions | $78.3 \pm 0.5$ | $68.6 \pm 0.2$ | $64.7 \pm 1.0$ | $50.3 \pm 0.5$ |
| w/o Molecular Features | $79.1 \pm 0.2$ | $67.1 \pm 0.4$ | $65.8 \pm 2.3$ | $51.7 \pm 0.6$ |

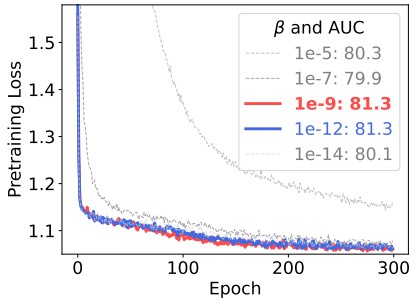 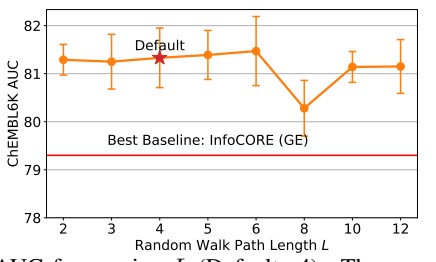

(a) Losses and AUC on varying $\beta$ (Default: 1e-9).  (b) AUC for varying $L$ (Default: 4). The error bar represents one standard deviation from ten runs.

Figure 5: Analysis on the hyperparameters: strength of prior $\beta$ and random walk length $L$. AUC is computed on the test set of ChEMBL2K.

in Table 4 cover all datasets. We observe that both cell morphology and gene expression features are crucial for achieving the best performance. Different biological targets have varying impacts across datasets: molecular structure has more influence on Broad6K and Biogen3K, while gene expression is more important for ChEMBL2K and ToxCast. Further studies on the cell morphology and gene expression are in appendix D.4.

### 6.3.2 HYPERPARAMETER ANALYSIS

Lastly, we perform analysis for the hyperparameters: the strength of the regularization to the prior Gaussian distribution $\beta$ and the length of the random walk paths $L$. Results are presented in Figure 5. We observe a trade-off between the principles of minimality and sufficiency in Figure 5a: a too-high $\beta$ value (minimal information) makes it challenging for the representation to be sufficiently expressive for molecular, gene expression, and cell morphology features, potentially degrading downstream performance. Conversely, a too-low $\beta$ value weakens minimality and may impair generalization. The convergence of the pretraining loss could serve as a good indicator to balance these aspects. For the hyperparameter $L$, we observe in Figure 5b that downstream performance on ChEMBL2K is relatively robust across a wide range of walk lengths. Further analysis of the random walk sampling is provided in appendix D.5.

## 7 CONCLUSION

In this work, we proposed learning molecular representations in a cell context with three modalities: molecular structure, gene expression, and cell morphology. We introduced the information bottleneck approach, InfoAlign, using a molecular graph encoder and multiple MLP decoders. InfoAlign learned minimal sufficient molecular representations extracted by reconstructing features in the random walk path on a cellular context graph. This context graph incorporated molecules, cell morphology, and gene expression information defined in scalar or vector spaces to construct nodes, and used various chemical, biological, and computational criteria to define their weighted edges. We demonstrated the theoretical and empirical advantages of the proposed method. InfoAlign outperformed other representation learning methods in various molecular property prediction and zero-shot molecule-morphology matching tasks.

ACKNOWLEDGMENTS

This study was supported by National Institutes of Health (R35 GM122547 to AEC) and an internship funded by the Massachusetts Life Sciences Center (to GL). This work was supported by NSF IIS-2142827, IIS-2146761, IIS-2234058, CBET-2332270, and ONR N00014-22-1-2507.

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

## A  MORE RELATED WORK ON REPRESENTATION LEARNING WITH DIFFERENT MODALITIES

Existing methods on multimodal alignment, such as CLIP (Radford et al., 2021), primarily address pairwise relationships between texts and images and use methods like InfoNCE (Oord et al., 2018; Wang et al., 2023a; Sanchez-Fernandez et al., 2023). These approaches use separate encoders for different modalities to compute contrastive loss, which is upper bounded by the number of negative examples (Poole et al., 2019). Subsequent research on molecules similarly focuses on pairwise alignment between molecules and cell images (Sanchez-Fernandez et al., 2023; Wang et al., 2023a), molecules and protein sequences (Huang et al., 2021), and molecules and text (Edwards et al., 2022; Jin et al., 2023). Although BioBridge (Wang et al., 2023b) handles multiple modalities, it leverages a knowledge graph for transforming representations between modalities rather than optimizing molecular representations.

## B  PROOF DETAILS

### B.1  PROOF OF EQ. (2)

Approximating the mutual information of high-dimensional variables is a challenging task (Gowri et al., 2024). For the input, latent, and target variables $X$, $Z$, and $Y$, the exact computation of the mutual information (MI) $I(Z;Y)$ and $I(X;Z)$ is intractable. For the molecule $x$, its latent representation $z$, and any biological target from cellular responses $y$, we introduce variational approximations $q(y|z)$ to obtain a lower bound on $I(Z;Y)$:

$$
\begin{aligned}
I(Z;Y) &= \mathbb{E}_{p(z,y)} \left[ \log \frac{p(z,y)q(y|z)}{p(y)p(z)q(y|z)} \right], \\
&= \mathbb{E}_{p(z,y)} \left[ \log \frac{q(y|z)}{p(y)} \right] + \mathbb{E}_{p(z)} \left[ \mathrm{KL} \left( p(y|z) \| q(y|z) \right) \right], \\
&\geq \mathbb{E}_{p(z,y)} \left[ \log q(y|z) \right] + H(Y) \triangleq I_{DLB}
\end{aligned}
\tag{5}
$$

This is because that $\mathrm{KL}\left(p(y|z)\|q(y|z)\right) \geq 0$. We introduce the variational approximations $q(z)$ for a upper bound on $I(X;Z)$:

$$
\begin{aligned}
I(X;Z) &= \mathbb{E}_{p(x,z)} \left[ \log \frac{p(x,z)q(z)}{p(x)p(z)q(z)} \right], \\
&= \mathbb{E}_{p(x,z)} \left[ \log \frac{p(z|x)}{q(z)} \right] - \mathbb{E}_{p(z)} \left[ (p(z)\|q(z)) \right], \\
&\leq \mathbb{E}_{p(x)} \left[ \mathrm{KL} \left( p(z|x) \| q(z) \right) \right] \triangleq I_{EUB}
\end{aligned}
\tag{6}
$$

### B.2  PROOF OF PROPOSITION 4.1

For the molecule $x$, its latent representation $z$, and any biological target from cellular responses $y$, we use the neural network parameterized critic $h(z,y)$ with the energy-based variational family for density approximation (Poole et al., 2019):

$$
q(y|z) = \frac{p(y)}{\mathbb{E}_{p(y)}\left[e^{h(z,y)}\right]} e^{h(z,y)}.
$$

Thus, we can rewrite $I_{DLB}$ based on the unnormalized distribution of $q(y|z)$:

$$
\begin{aligned}
I_{DLB} &= \mathbb{E}_{p(z,y)} \left[ \log q(y|z) \right] + H(Y) \\
&= \mathbb{E}_{p(z,y)} \left[ \log \left( \frac{p(y)}{\mathbb{E}_{p(y)}\left[e^{h(z,y)}\right]} e^{h(z,y)} \right) \right] - \mathbb{E}_{p(y)} \left[ \log p(y) \right], \\
&= \mathbb{E}_{p(z,y)} \left[ h(z,y) \right] - \mathbb{E}_{p(z,y)} \left[ \mathbb{E}_{p(y)}[e^{h(z,y)}] \right], \\
&= \mathbb{E}_{p(z,y)} \left[ h(z,y) \right] - \mathbb{E}_{p(z)} (\log \tilde{Z}(z)),
\end{aligned}
\tag{7}
$$

where $\tilde{Z}(z) = \mathbb{E}_{p(y)}[e^{h(z,y)}]$ is the partition function.

Note that the log partition function is intractable. Poole et al. (2019) introduced a new variational parameter $a(\cdot)$ to upper bound $\tilde{Z}(z)$, deriving a tractable lower bound for $I_{DLB}$:

$$I_{DLB} \geq \mathbb{E}_{p(z,y)}[h(z,y)] - \mathbb{E}_{p(z)}\left[\frac{\mathbb{E}_{p(y)}[e^{h(z,y)}]}{a(z)} + \log(a(z)) - 1\right]. \tag{8}$$

This is because $\forall x, a > 0$, the inequality $\log(x) \leq \frac{x}{a} + \log(a) - 1$ holds, which can be applied to the second term of Eq. (7). The $I_{NWJ}$ bound (Nguyen et al., 2010) is a special case where $a(z) = e$.

$$I_{NWJ} \triangleq \mathbb{E}_{p(z,y)}[h(z,y)] - \mathbb{E}_{p(z)}\left[\frac{\mathbb{E}_{p(y)}[e^{h(z,y)}]}{e} + \log(e) - 1\right]$$

$$= \mathbb{E}_{p(z,y)}[h(z,y)] - e^{-1}\mathbb{E}_{p(z)}[\tilde{Z}(z)]. \tag{9}$$

$I_{NWJ}$ has high variance due to the estimation of the upper bound on the log partition function. Based on $I_{NWJ}$ and multiple examples, one can derive the encoder-based lower bound $I_{ELB}$ for InfoNCE.

Suppose there are $K - 1$ additional examples independently and identically sampled and denoted as $y_{2:K}$, and the critic is configured with parameters $a(\cdot)$ as $1 + \log\frac{e^{h(z,y)}}{a(z;y,y_{2:K})}$. Then, we can rewrite $I_{NWJ}$ for its multi-sample version:

$$I_{NWJ} = \mathbb{E}_{p(z,y)p(y_{2:K})}\left[1 + \log\frac{e^{h(z,y)}}{a(z;y,y_{2:K})}\right] - e^{-1}\mathbb{E}_{p(y)p(z)p(y_{2:K})}\left[e^{1+\log\frac{e^{h(z,y)}}{a(z;y,y_{2:K})}}\right],$$

$$= 1 + \mathbb{E}_{p(z,y)p(y_{2:K})}\left[\log\frac{e^{h(z,y)}}{a(z;y,y_{2:K})}\right] - \mathbb{E}_{p(y)p(y_{2:K})p(z)}\left[\frac{e^{h(z,y)}}{a(z;y,y_{2:K})}\right]. \tag{10}$$

Multiple samples can be utilized for the Monte Carlo method $m(z;y,y_{2:K})$ to estimate the upper bound on the partition function $a(z;y,y_{2:K})$:

$$a(z;y,y_{2:K}) = m(z;y,y_{2:K}) = \frac{1}{K}\left(e^{h(z,y)} + \sum_{i=2}^{K} e^{h(z,y_i)}\right),$$

where $K - 1$ independent samples are drawn from $\prod_i p(y_i)$ and one sample from $p(z,y)$ for the term $\mathbb{E}_{p(z,y)p(y_{2:K})}[\cdot]$ or $K$ samples from $\prod_{i=1}^{K} p(y_i)$ (we set $y_1 = y$) for a $p(z)$ sample in the $\mathbb{E}_{p(y)p(y_{2:K})}[\cdot]$ term. Therefore, we can derive $I_{ELB} \triangleq I_{NCE}$:

$$I_{ELB} \triangleq I_{NCE} = 1 + \mathbb{E}_{p(z,y)p(y_{2:K})}\left[\log\frac{e^{h(z,y)}}{m(z;y,y_{2:K})}\right] - \mathbb{E}_{p(y)p(y_{2:K})p(z)}\left[\frac{e^{h(z,y)}}{m(z;y,y_{2:K})}\right],$$

$$= 1 + \mathbb{E}_{p(Y|Z)p(z)p(y_{2:K})}\left[\log\frac{e^{h(z,y)}}{\frac{1}{K}\sum_{i=1}^{K} e^{h(z,y_i)}}\right]$$

$$- \mathbb{E}_{p(y)p(y_{2:K})p(z)}\left[\frac{e^{h(z,y)}}{\frac{1}{K}\sum_{i=1}^{K} e^{h(z,y_i)}}\right],$$

$$= \mathbb{E}_{p(z,y)}[h(z,y)] - \mathbb{E}_{p(z)}\left[\log\frac{1}{K}\sum_{i=1}^{K} e^{h(z,y_i)}\right]. \tag{11}$$

Note that for $\mathbb{E}_{p(y)p(y_{2:K})p(z)}[\cdot]$, we average the bound over $K$ replicates as well to ensure that the last term in Eq. (10) is the constant 1. Now, $I_{ELB}$ or $I_{NCE}$ is upper bounded by $\log K$, rather than $a(\cdot)$. Hence, the difference between $I_{DLB}$ and $I_{ELB}$ is

$$I_{DLB} - I_{ELB} = \mathbb{E}_{p(z)}\left[\log\frac{1}{K}\sum_{i=1}^{K} e^{h(z,y_i)}\right] - \mathbb{E}_{p(z)}(\log\tilde{Z}(z)) \geq 0. \tag{12}$$

When $K$ is sufficiently large to estimate the partition function, we have $\mathbb{E}_{p(z)}\left[\log\left(\mathbb{E}_{p(y)}\left[e^{h(z,y)}\right]\right)\right]$ for the left term, indicating that $I_{DLB} - I_{ELB} = 0$. Since $I_{NCE}$ is upper bounded by $\log K$ (Oord et al., 2018), smaller values of $K$ may result in a less tight $I_{ELB}$, causing $I_{DLB} - I_{ELB} \geq 0$ to always hold. In particular, $I(Z;Y) > \log K$ implies that the bound $I_{ELB}$ will be loose.

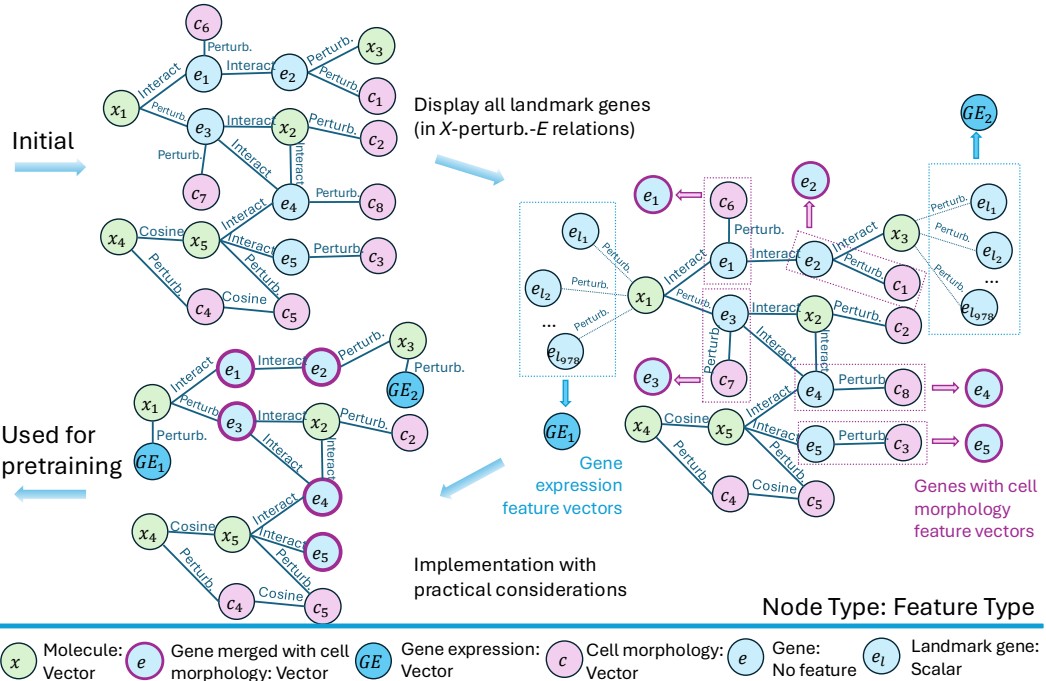

Figure 6: From the initial idea in Section 4 to the practical implementation of the context graph, we first display relations between molecules and all the landmark genes from Wang et al. (2016) for the $X_1 - E_3$ and $X_3 - E_2$ relationships. $E_3$ and $E_2$ are landmark genes involved in small molecule perturbations and cell morphology perturbation; we display them separately for clarity. Next, we merge all landmark genes into new gene expression nodes and integrate genes from genetic perturbations in the JUMP dataset (Chandrasekaran et al., 2023) with cell morphology features. Practical considerations are detailed in Section 5 and appendix C.

## C  CONTEXT GRAPH AND MODEL DETAILS

### C.1  EDGE CONSTRUCTION

Edges represent similarity relationships between molecules, genes and cells. According to the chemical or biological criteria, we have following types of edges:

1. Molecule-Cell Morphology Edges: These edges are introduced through molecule perturbation experiments from cell painting datasets created by Bray et al. (2016) and the JUMP dataset (Chandrasekaran et al., 2023). It links molecule nodes with cell morphology nodes. We use the edge weight 1 for all these edges.

2. Edge-Cell Morphology Edges: These edges should be introduced by genetic perturbation from the JUMP dataset Chandrasekaran et al. (2023). The perturbations are either based on gene overexpression (ORF) or gene knockout techniques (CRISPR). They link the gene nodes and the cell morphology nodes. However, the genes introduced by the genetic perturbations lack gene expression profiling from (Subramanian et al., 2017) as node features. We did not implement gene-cell morphology edges from (Chandrasekaran et al., 2023) due to the absence of differential gene expression profiling values (Subramanian et al., 2017). Instead, we merged the gene nodes from (Chandrasekaran et al., 2023) with their linked cell morphology nodes, creating single nodes. This approach enables a more efficient context graph, incorporating some gene nodes with cell morphology features.

3. Molecule-Gene Edges: These edges could represent molecule-gene binding and regulation relationships, linking molecules to genes (Himmelstein et al., 2017). Some links can be sourced from (Himmelstein et al., 2017), and we also retrieve gene-molecule links from (Wang et al., 2016) by selecting the top 5% absolute differential expression values.

    4. Gene-Gene Edges: These edges denote the relationships of gene-gene covariance and interaction and we use the links from (Himmelstein et al., 2017).

We enrich the edges in the context graph by incorporating computational similarity edges, where cosine similarity is computed among within nodes having the same type and feature vectors. We note that the cell morphology features from (Bray et al., 2016) and (Chandrasekaran et al., 2023) have different dimensions since the latter has applied batch correction techniques Arevalo et al. (2023) on the CellProfiler features (Carpenter et al., 2006). Thus, we cannot compute the similarity between these two subsets of cell morphology nodes. We use (1) a 0.8 similarity threshold and (2) a minimal sparsity of 99.5% by selecting top 0.5% similar edges to avoid excessive noise in computational similar edges.

### C.2 DATASET SOURCES OF NODES

Here are the datasets we used to create different types of nodes on the context graph:

- Molecule nodes: Molecular nodes are sourced from two cell painting datasets: one by Bray et al. (2017) and the other from the recently released JUMP dataset (Chandrasekaran et al., 2023), and the third source from Wang et al. (2016), which are used to study adverse drug reactions.

- Gene nodes: Gene nodes are from the landmark genes used by Wang et al. (2016) in creating the LINCS L1000 profiling of drugs. Other gene nodes come from genetic perturbations in the JUMP dataset (Chandrasekaran et al., 2023). The gene nodes from (Chandrasekaran et al., 2023) have cell morphology features as described in appendix C.1. The landmark gene nodes from (Wang et al., 2016) have scalar gene expression profiles, but these values are updated in the new gene expression nodes.

- Cell morphology nodes: Cell nodes are sourced from the two cell painting datasets (Bray et al., 2016; Chandrasekaran et al., 2023).

- Gene expression nodes: Based on landmark genes from (Wang et al., 2016), each gene expression node summarizes all gene expression profiles into vectors from a small molecule perturbation. Since Wang et al. (2016) measured the same landmark genes for a set of molecules, we update new gene expression nodes with feature vectors for all these landmark genes. This approach efficiently constructs decoding targets from molecules to gene expression profiles and prevents redundant gene-molecule connections.

We present an example of the cellular context graph in Figure 6.

### C.3 MODEL DETAILS

We use a five-layer GIN (Xu et al., 2019), following previous GNN pre-training work (Hu et al., 2020a). The hidden dimension is set to 300, with a batch normalization layer and summation used for readout of the graph representation from node-level features.

For the MLP decoders, they have three layers. The input dimension is 300, with a hidden dimension of 1200. The output dimension matches the corresponding node features from the context graph in pretraining. For the newly introduced hyperparameters $\beta = 1e - 9$ and $L = 4$, details on their studies can be found in Section 6.3.2.

In downstream tasks, the GNN encoder is frozen, and new MLP decoders, which output task predictions, are applied to the molecular representation from the GNN encoder.

## D EXPERIMENT DETAILS

### D.1 PREDICTION DATASETS

All experiments were run on a single 32G V100. Molecules can interact with multiple cells and genes to generate cell morphology and gene expression features. These molecules act as perturbations, with the resulting cell states measured as gene expression values for thousands of genes (Subramanian

Table 5: Datasets and task information. Classf. denotes classification and Regr. denotes regression.

| Dataset | Type | # Task | # Molecules | # Atoms Avg./Max | # Edges Avg./Max | # Available Cell Morphology | # Available Gene Expressions |
|---|---|---|---|---|---|---|---|
| ChEMBL2K | Classf. | 41 | 2355 | 23.7/61 | 25.6/68 | 2353 | 631 |
| Broad6K | Classf. | 32 | 6567 | 34.1/74 | 36.8/82 | 2673 | 1138 |
| ToxCast | Classf. | 617 | 8576 | 18.8/124 | 19.3/134 | N.A. | N.A. |
| Biogen3K | Regr. | 6 | 3521 | 23.2/78 | 25.3/84 | N.A. | N.A. |

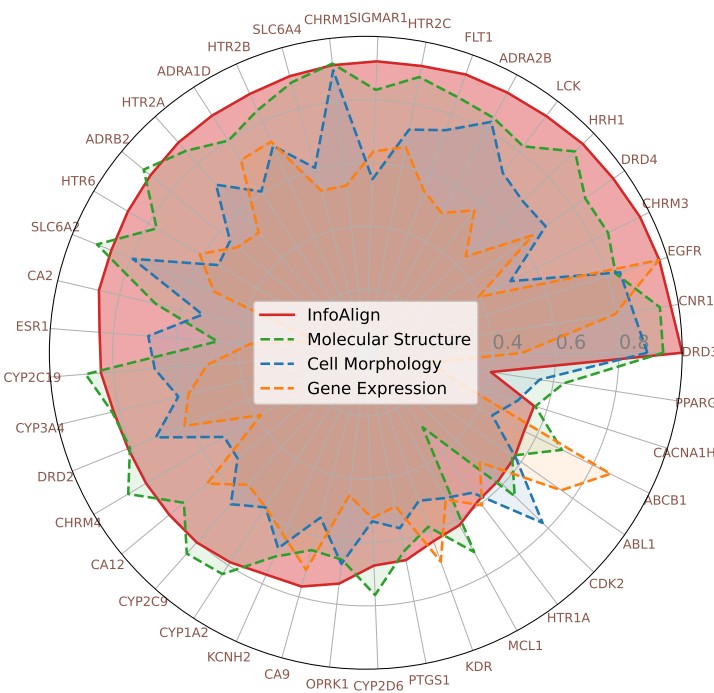

Figure 7: An overview of the representation's predictive performance on all 41 bioactivity prediction tasks in ChEMBL2K. Results for molecular structure are obtained from the best method ContextPred. Results for cell morphology and gene expression come from the best method based on MLPs.

et al., 2017) and/or as morphology features from Cell Painting microscopy images (Chandrasekaran et al., 2023; Cimini et al., 2023). Due to the high cost and complexity of these experiments, not all downstream datasets include morphology or gene expression data for each molecule. Notably, ToxCast and Biogen3K lack these features. Prediction dataset statistics are detailed in Table 5.

- **ChEMBL2K (Gaulton et al., 2012):** The dataset is a subset of the ChEMBL dataset (Gaulton et al., 2012), overlapping with the JUMP CP (Chandrasekaran et al., 2023) datasets. We determined activity using the "activity_comment" provided by ChEMBL. If not, we applied a threshold of 6.5, labeling compounds with pChEMBL > 6.5 as active. We exclude all molecules in the dataset from the pretraining set to avoid data leakage. There are a total of 41 tasks related to protein binding affinity, which are converted to binary activity values. We filter the dataset to ensure that each task has at least one positive and five negative examples.

- **Broad6K (Moshkov et al., 2023):** The original version provided by Moshkov et al. (2023) is a collection of 16,170 molecules tested in 270 assays, resulting in a total of 585,439 readouts. However, there are a large number of missing values, with 153 assays having a missing value percentage above 99%. To mitigate bias in the conclusions, we extract subsets where the percentage is less than 50%.

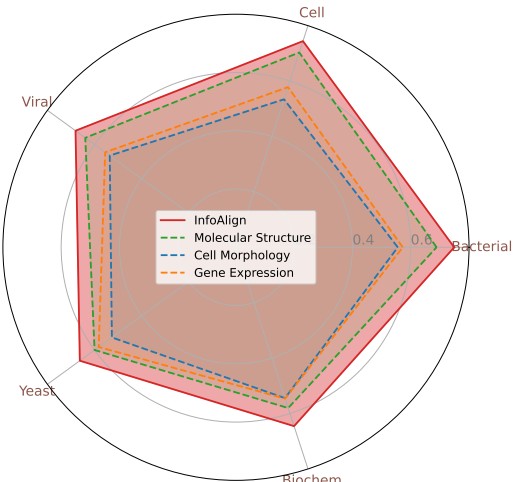

Figure 8: An overview of the representation's predictive performance across five major task categories on Broad6K. Results for molecular structure are obtained from the best method based on JOAO. Results for cell morphology come from the best method based on RF. Results for gene expression are derived from the best method based on MLP.

- ToxCast (Richard et al., 2016): The toxicology data is collected from the "Toxicology in the 21st Century" initiative, widely utilized in many graph machine learning models (Hu et al., 2020b). The dataset comprises 8,576 molecules and 617 binary classification tasks.

- Biogen3K (Fang et al., 2023): The dataset includes properties that describe the disposition of a drug in the body, including absorption, distribution, metabolism, and excretion (ADME). It is collected from 120 Biogen datasets across six ADME in vitro endpoints over 20 time points spanning about 2 years. The endpoints include human liver microsomal (HLM) stability reported as intrinsic clearance (Clint, mL/min/kg), MDR1-MDCK efflux ratio (ER), Solubility at pH 6.8 (µg/mL), rat liver microsomal (RLM) stability reported as intrinsic clearance (Clint, mL/min/kg), human plasma protein binding (hPPB) percent unbound, and rat plasma protein binding (rPPB) percent unbound.

We utilize scaffold-splitting with a ratio of 0.6:0.15:0.25 for all datasets. We use the Area under the curve (AUC) score for classification and mean absolute error (MAE) for regression. We report the mean and standard deviations from ten runs.

## D.2 IMPLEMENTATION AND BASELINE

We consider baselines from three representation sources: molecular structures, cell morphology, and gene expressions. Moreover, we have three different ways to represent molecular structures, including fingerprints based on domain knowledge, GNNs based on the graph structure of molecules, and chemical language models (ChemLM) based on SMILES-sequence structure of molecules.

1. Molecular descriptors/fingerprints (Rogers & Hahn, 2010) (Structure only): We train MLPs, Random Forests (RF), and Gaussian Processes (GP) on these representations.

2. Pretrained GNN representations (Hu et al., 2020a) (Structure only): We consider AttrMask, ContextPred, and EdgePred with supervised pretraining (Hu et al., 2020a). We also include GraphCL (You et al., 2020), GROVER (Rong et al., 2020), JOAO (You et al., 2021), MGSSL (Zhang et al., 2021), GraphLoG (Xu et al., 2021), GraphMAE (Hou et al., 2022), DSLA (Kim et al., 2022), and UniMol (Zhou et al., 2023). We implement GraphCL, GROVER, and JOAO based on (Wang et al., 2024). Fine-tuned MLPs are applied on top of the pretrained representations.

3. Pretrained ChemLM representations (Frey et al., 2023) (Structure only): We consider pretrained models such as 102M Roberta and 87M GPT2 implemented by (Mary et al.,

Table 6: Ablation studies on the pretraining loss. Different node features are removed from the context graph to assess their impact on downstream tasks. Avg. AUC is reported.

|  | ChEMBL2K AUC ↑ | Broad6K AUC ↑ | ToxCast AUC ↑ | Biogen3K AUC ↑ |
|---|---|---|---|---|
| InfoAlign | $81.33 \pm 0.62$ | $69.95 \pm 0.09$ | $66.36 \pm 1.05$ | $49.42 \pm 0.18$ |
| w/o Cell-Related Nodes | $79.57 \pm 0.58$ | $68.41 \pm 0.31$ | $65.11 \pm 0.82$ | $51.21 \pm 0.17$ |
| w/o Gene-Related Nodes | $77.97 \pm 0.33$ | $67.10 \pm 0.17$ | $64.93 \pm 0.96$ | $51.57 \pm 0.46$ |
| w/o Cell-Related Loss | $80.70 \pm 0.60$ | $68.60 \pm 0.10$ | $65.50 \pm 1.10$ | $51.70 \pm 1.10$ |
| w/o Gene-Related Loss | $78.30 \pm 0.50$ | $68.60 \pm 0.20$ | $64.70 \pm 1.00$ | $50.30 \pm 0.50$ |

2024). We also include MolT5 (Edwards et al., 2022) and 19M ChemGPT (Frey et al., 2023). We apply fine-tuned MLPs on top of these pretrained representations.

4. Cell Morphology (Rogers & Hahn, 2010) (Cell or Structure only): Cell morphology features are available in for part of molecules in the ChEMBL2K and Broad6K datasets. We train MLPs, RF, and GP on these representations. Note that not all molecules have corresponding cell morphology feature vectors; in such cases, we replace the predictions on the missing feature with ML predictions on the structure.

5. Gene Expression (Rogers & Hahn, 2010) (Gene or Structure only): Differential gene expression values are available for part of molecules in the ChEMBL2K and Broad6K datasets. We train MLPs, RF, and GP on these representations. Note that not all molecules have corresponding gene expression vectors over landmark genes; in such cases, we replace the predictions on the missing feature with ML predictions on the structure.

6. CLOOME (Sanchez-Fernandez et al., 2023) and InfoCORE (Wang et al., 2023a) (Structure-Cell or Structure-Gene aligned): CLOOME utilizes ResNet (He et al., 2016) and descriptor-based MLP to align representation from cell morphology images with the molecular structure representation. We use their pretrained MLP to obtain molecular representations and fine-tune another MLP on top of these representations. InfoCORE has two versions, InfoCORE-CP and InfoCORE-GE, which align the molecular graph representation with cell morphology features or differential gene expression features, respectively. We use both versions as baselines and fine-tune another MLP on top of these representations.

## D.3   MORE RESULTS FOR MOLECULAR PROPERTY PREDICTION

We present additional comparisons on the ChEMBL2K dataset between basic representation approaches and InfoAlign across all task dimensions in Figure 7. Similarly, results for the Broad6K dataset, comparing basic representations across five major task dimensions (Cell, Yeast, Viral, Biochem, and Bacterial-related targets), are shown in Figure 8. Combined with Tables 1 and 2, these detailed results lead to further observations:

(1) Different structure-based molecular representations vary in sensitivity to model architecture. Dramatic performance drops occur with Morgan FP when replacing the MLP architecture with RF or GP in the ChEMBL2K and Broad6K datasets. Conversely, in the Biogen3K dataset, RF and GP significantly outperform MLP. In contrast, pretrained GNN and ChemLM representations maintain more consistent performance across various datasets.

(2) Learning universal molecular representations solely from molecular structures remains challenging, even within the representation category. For pretrained GNN representations, ContextPred outperforms others on the ChEMBL2K dataset. JOAO excels on the Broad6K dataset. UniMol and GraphMAE are the best pretrained GNN representations on ToxCast and Biogen3K datasets, respectively. For ChemLM representations, MolT5 excels over other sequential-based models in the ToxCast and Biogen3K datasets, but this is not the case with the ChEMBL2K and Broad6K datasets. Different datasets may emphasize varied aspects of bioactivity classification or regression and pose generalization challenges for molecular representation learning.

(3) InfoAlign shows strong generalization for the targets of non-human cells, as shown in Figure 8. Although the context graph primarily uses data from small molecule and genetic perturbation datasets (Bray et al., 2016; Chandrasekaran et al., 2023) focused on human cell cultures, InfoAlign

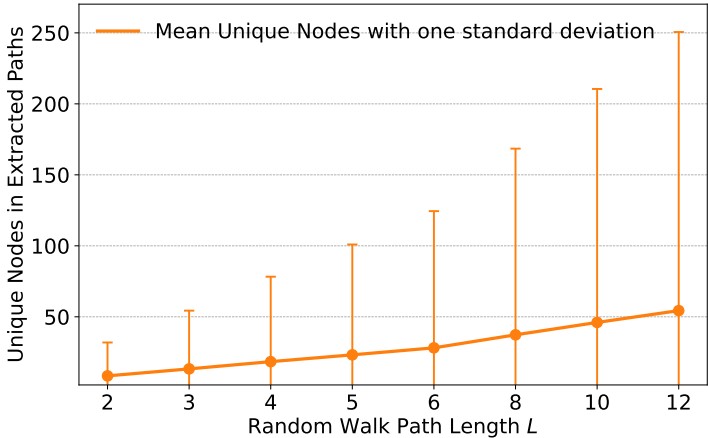

Figure 9: The mean, with one standard deviation, of the number of unique nodes extracted from the random walk paths. We vary the walk length $L$ from 2 to 12, showing that random walks can sample diverse neighbors, with the number exceeding the walk length.

also exhibits robust generalization to bacterial and viral targets compared to basic representation approaches.

Table 7: Performance comparison of fully fine-tuned UniMol and InfoAlign models with their representations across various datasets. "Others" refers to cell morphology and gene expression features.

| Fine-tuning Method | ChEMBL2K | Broad6K | ToxCast | Biogen3K |
|---|---|---|---|---|
| Representatio from UniMol | 76.8±0.4 | 65.4±0.1 | 64.6±0.2 | 55.8±2.8 |
| Representation from UniMol and Others | 77.5±0.1 | 66.4±0.5 | NA | NA |
| Fully-tuned UniMol | 78.9±0.2 | 65.1±1.0 | 71.3±0.6 | 43.6±0.3 |
| Representation from InfoAlign | 81.3±0.6 | 70.0±0.1 | 66.4±1.1 | 49.4±0.2 |
| Fully-tuned InfoAlign | 80.1±0.9 | 69.2±0.7 | 72.0±0.5 | 42.8±1.1 |

Additionally, we set up further comparisons, including: (1) the concatenated representation from UniMol (Zhou et al., 2023) using cell morphology and gene expression on the ChEMBL2K and Broad6K datasets, and (2) fully fine-tuned InfoAlign and UniMol, including both the representation encoder and MLP predictors.

Results are presented in Table 7. First, we observe that concatenating UniMol representations with cell morphology and gene expression features improves performance in prediction tasks, though it still does not match the performance of InfoAlign. InfoAlign achieves the best results by aligning molecular representations with these features during pretraining, rather than in the downstream stage.

Second, we observe that fully fine-tuning benefits both UniMol and InfoAlign on ToxCast and Biogen3K datasets. While fully fine-tuning improves UniMol representation on ChEMBL, InfoAlign, with only MLP decoder tuning, achieves the best performance. On Broad6K, fully fine-tuning is less effective for both InfoAlign and UniMol compared to tuning only the MLP. These results suggest that, if resources allow, fully fine-tuning should be preferred for better overall performance, especially for UniMol, which requires more time and resources due to the use of 3D molecular structures. If resources are limited, InfoAlign's representation provides a strong alternative without the need for full fine-tuning.

### D.4 MORE ABLATION STUDIES ON CELL MORPHOLOGY AND GENE EXPRESSION

We also conducted ablation studies to assess the importance of cell morphology and gene expression in constructing the context graph. We removed either all cell morphology-related nodes or gene

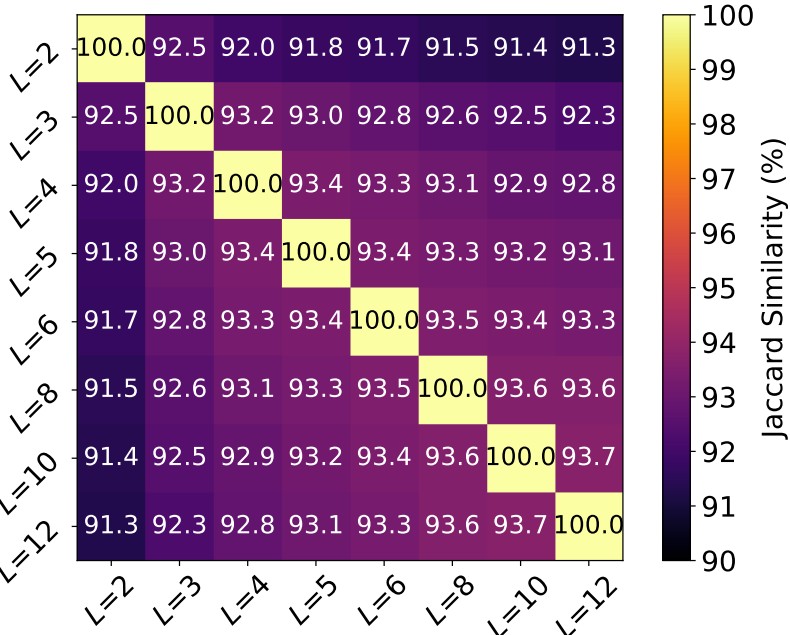

Figure 10: Jaccard Similarity of Neighborhoods Across Different Walk Lengths: We calculate the Jaccard similarity of neighborhoods from different walk lengths. We compare pairwise similarities for the same molecule across varying walk lengths $L$ in the pretraining set and report the average similarity for the set.

expression-related nodes from the context graph and pretrained InfoAlign. Combined with results from Table 4 (without cell/gene-related loss terms), additional results where nodes are removed from the context graph are presented in Table 6.

First, we confirm the importance of cell morphology and gene expression as nodes in the context graph for pretraining. Second, we observe a greater performance drop when removing nodes compared to excluding the loss terms. This highlights the importance of including diverse data types as nodes, even without features, and suggests a promising direction for improving InfoAlign's pretraining by enhancing the context graph with virtual nodes.

## D.5    MORE PERFORMANCE ANALYSIS ON THE RANDOM WALK

We conduct more experiments to analyze how random walk helps InfoAlign pretraining with diverse neighbor sampling. We cache the random walk results for 100 epochs and study the number of unique nodes at varying walk lengths. In Figure 9, we report the mean and standard deviation of the number of unique nodes for all molecules in the pretraining set at each walk length. We observe that the number of unique nodes is larger and varies compared to the corresponding walk length, indicating that diverse neighbors are sampled in different training steps.

We further explore the Jaccard similarity of neighborhoods extracted for the same molecule across varying walk paths, averaging similarity scores over all pretraining molecules. The pairwise similarities for different walk lengths are shown in Figure 10. We observe that similarity decreases as the difference in walk lengths increases, but remains above 90%. This suggests that while random walks sample diverse neighbors, varying the walk length does not significantly affect the neighborhood, which may explain the stable performance of InfoAlign in Figure 5b. These results also show that even with a walk length of 2, diverse neighbors can be obtained, likely due to high-degree nodes in the context graph.

