# OpenReview forum: "Learning Molecular Representation in a Cell"
_ICLR.cc/2025/Conference — ICLR 2025 Poster_

### Official Review · Reviewer_Sxqu · 2024-10-23

**Soundness:** 3
**Presentation:** 2
**Contribution:** 3
**Rating:** 3
**Confidence:** 4

**Summary:**

This article proposes a pre-training method where the authors utilize genes, cells, and molecular information as nodes, and construct a context graph using the interactive or similar relationships between different components as edges. The authors then extract paths on the graph using random walks to construct training data, design training objectives from the perspective of mutual information, and integrate cellular and genetic information into molecular representations. After pre-training, the authors fine-tune the model on downstream tasks across four datasets to validate its performance.

**Strengths:**

- This article pioneeringly incorporates cellular modality information in molecular representation learning, which enables the learned molecular representations to achieve better results in biochemistry-related tasks.
- The article conducts sufficient experiments to validate the effectiveness of the model.
- The paper is well-motivated; adding cellular modality data to enhance the model's performance in biochemical aspects is very reasonable.

**Weaknesses:**

- The writing requires improvement; the methodology is difficult to follow, and many variables are not clearly defined. See Questions for details.
- The description of the dataset and downstream tasks is not sufficiently detailed, with some parts being confusing. See Questions for details.

**Questions:**

- **Methodology**
  - How is the neighboring node of $x$ defined in line 250? Is it the neighbor of $x$ on the context graph or all the $v_i$ on the sampled path?
  - As shown in Fig. 2, different decoders are used to reconstruct the features of different types of nodes. However, if there are multiple neighbors $v_1, v_2, \ldots, v_n$ of a certain molecule $x$ sharing the same type, how can the decoder decode various $y_{v_i}$ from the same encoded latent  $z$ of $x$ without any additional information about $v_i$ being provided?

- **Experiments**
  - The ChEMBL dataset provides information on whether a certain molecule exhibits activity against a specific target. In this manuscript, a task is defined as predicting whether a molecule can interact with a given target. However, it is unclear why the molecule would be characterized by **Cell Morphology** and **Gene Expression**. Such information should be related to the task (target) itself rather than the input molecule. Could the authors explain how **Gene Expression** and **Cell Morphology** data are generated for molecules?
  - To substantiate the claim that the proposed multimodal alignment method more effectively models molecular properties within a cellular context, the authors must include a baseline comparison using a simple concatenation approach for alignment. Specifically, the authors should employ a pretrained GNN, such as Uni-Mol[1], to extract molecular features and concatenate them with Cell Morphology and Gene Expression representations as inputs for prediction tasks on the ChEMBL2k and Broad6k datasets.
  - During the fine-tuning for downstream tasks, is the encoder frozen or is a full-fine-tune performed with the encoder also being trained?

**If the authors can address all my major concerns, I would be pleased to raise the score.**

[1] Zhou G, Gao Z, Ding Q, et al. Uni-mol: A universal 3d molecular representation learning framework[J]. 2023.

---

> ### Author Response · Authors · 2024-11-18
> **Author Response**
>
> We sincerely appreciate the reviewer's thoughtful suggestions and questions. We have provided point-by-point answers to each weakness and question. We have also revised the main text and appendix to incorporate the reviewer's valuable feedback, with all changes clearly highlighted in blue for ease of reference. Should any concerns remain, we remain fully committed to addressing them promptly and thoroughly.
>
> ## W1, Q1: Definition of neighbors
>
> As stated in Eq (3) ($\sum_{v\in\mathcal{P}_x}$), the neighbor $v$ refers to the node in the random walk path. We have revised the text for better clarity.
>
> ## W1, Q2 Decoding for multiple neighbors
>
> Thanks for your question. If multiple neighbors share the same data type, such as cell morphology, the latent representation of the molecule is passed through a single decoder for that data type. Based on Eq. (3), the decoder's output is then aligned with the different features associated with $v_1, v_2, \dots, v_n$, with weights assigned according to $\alpha(v_i \mid \mathcal{P}_x)$ ($i=1, 2, \dots, n$).
>
> ## W2, Q3: Generation of gene expression and cell morphology
>
> Thanks for your question. Most explanations are provided in the Introduction and Related Work sections. We apologize for not referencing them when introducing the dataset and have revised the paper (Section 6.1.1, Appendix D.1) to address your concern. Below is a brief summary:
>
> Molecules act as perturbations that yield perturbed cell states; those cell states can be measured in two ways relevant here: as gene expression values for a thousand or more genes [1] and/or microscopy Cell Painting images [2], which are represented by a thousand or more morphology features [3]. This is how a gene expression and morphology profile are associated with a given molecule.
>
> Generating cell morphology and gene expression features requires extensive and costly experiments, so downstream tasks like ToxCast and Biogen3K may not always include these features.
> As described in Appendix D.1, ChEMBL2K is a subset overlapping with the existing JUMP-CP dataset [4], the largest cell painting dataset for cell morphology features. Relevant gene expression data for the molecules in ChEMBL2K are sourced from [6]. The available data for both types are detailed in Table 5 in the appendix.
>
> ## W2, Q4: Feature concatenation
>
> Thanks for your comments. We appreciate the helpful analysis and have now included it in Appendix C.3 and Table 7. The requested new results are provided in the table below:
>
> | Dataset    | UniMol     | UniMol + Other Features  | InfoAlign   |
> |------------|------------|--------------------------|-------------|
> | ChEMBL2K   | 76.8±0.4   | 77.51±0.08               | 81.3±0.6    |
> | Broad6K    | 65.4±0.1   | 66.43±0.49               | 70.0±0.1    |
>
> We observe that concatenating UniMol representations with cell morphology and gene expression features improves performance in prediction tasks. However, it still does not match the performance of InfoAlign, which achieves the best results by aligning molecular representations with cell morphology and gene expression features during pretraining, rather than in the downstream stage.
>
> ## W2, Q4: Fine-tuning
>
> Thanks for your comment. For all pre-trained methods, the encoder is frozen during fine-tuning, and only the MLP is trained for prediction. We have now explained this in Section 6.1 and Appendix C.3.
>
> ## Reference
>
> [1] A Next Generation Connectivity Map: L1000 Platform and the First 1,000,000 Profiles. Cell. 2017.
>
> [2] Three million images and morphological profiles of cells treated with matched chemical and genetic perturbations. Nature Method. 2024.
>
> [3] Optimizing the Cell Painting assay for image-based profiling. Nature Protocols. 2023.
>
> [4] JUMP Cell Painting dataset: morphological impact of 136,000 chemical and genetic perturbations.
>
> [5] Drug-induced adverse events prediction with the LINCS L1000 data. Bioinformatics. 2016.

---

> ### Comment · Reviewer_Sxqu · 2024-11-19
> **Further Questions**
>
> 1. **Decoding for Multiple Neighbors**
>
>    Can I consider the decoding process as the procedure of autoregressively generating the sampled paths?
>
> 2. **Finetuning**
>
>      It does not make sense to freeze the encoders as the original paper of UniMol[1] utilized full finetuning on its downstream tasks (some of the datasets even have less data than those used in the article), so do other molecular pretraining works such as [2]. More importantly, it does not make sense that the Morgan fingerprint outperforms most of the molecular pretraining models, as shown in Table 1 (even outperforming all on ChemBL2k), as this phenomenon challenges the meaning of the entire field of molecular pretraining. The performance of these molecular pretraining models has been severely underestimated. The authors should demonstrate the performance of the fully finetuned version and also update the results in response to "W2, Q4, feature concatenation".
>
>
> [1] Zhou G, Gao Z, Ding Q, et al. Uni-mol: A universal 3d molecular representation learning framework[J]. 2023.
>
> [2] Rong Y, Bian Y, Xu T, et al. Self-supervised graph transformer on large-scale molecular data[J]. Advances in neural information processing systems, 2020, 33: 12559-12571.

---

> > ### Author Response · Authors · 2024-11-19
> > **Author Response to Further Questions**
> >
> > We appreciate the reviewer’s prompt response and follow-up question. We are happy to provide further clarification on the concerns. Should any issues remain, we are committed to addressing them promptly and thoroughly.
> >
> > ## Decoding for Multiple Neighbors
> >
> > The decoding process predicts node features from the context graph. The values of these features range from 0 to 1 (Lines 292-293). This is not autoregressive generation. Intuitively, it resembles multi-label prediction. And the molecule may be aligned with different nodes simultaneously, each with different weights.
> >
> > ## Finetuning
> >
> > We appreciate your comment.  Let's answer them point-to-point.
> >
> > > It does not make sense to freeze the encoders as the original paper of UniMol[1] utilized full finetuning on its downstream tasks (some of the datasets even have less data than those used in the article), so do other molecular pretraining works such as [2].
> >
> > We freeze the encoder based on the following motivations:
> >
> > 1. **Task Setting**: As indicated in the title, our goal is "Learning Molecular Representation." We focus on evaluating the quality of representations from pretraining, rather than fine-tuned encoders. In this context, the core in Section 6.1 is the predictive ability of different molecular representations. The subsequent points (2 and 3) provide a fair and focused evaluation pipeline.
> >
> > 2. **Fair and Consistent Evaluation**: We freeze the encoder for **all** approaches (including InfoAlign) to ensure a fair and consistent comparison across methods.
> >
> > 3. **Occam's Razor for Evaluation**: We follow the principle of parsimony, which we believe is good practice for model evaluation. Freezing the encoder helps avoid introducing too many assumptions and better isolates the impact of the molecular representations themselves.
> >
> > 4. **Practicality**: Obtaining molecular representations has additional benefits. It is less resource-intensive than fully fine-tuning models for downstream applications. Notably, the UniMol model also provides an official API for accessing molecular representations, which suggests they support evaluation of their representations in a similar manner.
> >
> > _______
> >
> > > More importantly, it does not make sense that the Morgan fingerprint outperforms most of the molecular pretraining models, as shown in Table 1 (even outperforming all on ChemBL2k), as this phenomenon challenges the entire field of molecular pretraining's existence.
> >
> > ### We found that Morgan fingerprints do not challenge the existence/progress of molecular pretraining.
> >
> > Morgan fingerprints do not outperform pretrained GNN representations on Broad6K and ToxCast, where the best GNN-based representations show significant improvements compared to the best methods based on the fingerprints. On Biogen3K and ChEMBL2K, the best methods based on fingerprints also do not surpass the performance of the best pretrained GNN representations.
> >
> > ### Fingerprints could be good baselines to promote better molecular pretraining.
> >
> > (1) Defining universal self-supervised tasks from molecular structures alone is challenging, as discussed in our Introduction and recent publications [1]. Molecular pretraining often requires domain-specific knowledge, which is difficult to capture with manually designed tasks. (2) While fingerprints are a classic method, they are not a weak baseline and can perform well in certain tasks. This observation can also be found in previous work [2,3].
> >
> > _______
> >
> > >  The performance of these molecular pretraining models has been severely underestimated. The authors should demonstrate the performance of the fully finetuned version and also update the results in response to "W2, Q4, feature concatenation".
> >
> > We apologize for any confusion. Our main contribution and evaluation focus on molecular representations, and we chose to freeze the encoder during pretraining to maintain a focused analysis. While fully fine-tuning can unlock the full potential of molecular pretraining models, it may introduce additional assumptions that could influence the analysis. In designing the fine-tuning pipeline, we first ensured a consistent and fair evaluation. Then, following Occam's razor, we froze the encoder to better isolate the impact of the molecular representations themselves. We hope this clarifies our rationale and appreciate your understanding.
> >
> > ## Reference
> > [1] Does GNN Pretraining Help Molecular Representation? NeurIPS 2022.
> >
> > [2] Understanding the limitations of deep models for molecular property prediction: Insights and solutions. NeurIPS 2023.
> >
> > [3] Enhancing activity prediction models in drug discovery with the ability to understand human language. ICML 2023.

---

> > > ### Comment · Reviewer_Sxqu · 2024-11-19
> > >
> > > The provided references [2, 3] only demonstrate that fingerprints can indeed outperform simple GNNs such as GCN and GAT on some tasks, but they do not prove that fingerprints possess the capability to match pretrain-GNNs. Moreover, in terms of the magnitude of improvement, pretrain-GNNs can bring about a very significant enhancement [1], which is larger than the performance gap shown in [2]. Therefore, this does not constitute a valid reason for not comparing fully finetuned models.
> > >
> > > [1] Rong Y, Bian Y, Xu T, et al. Self-supervised graph transformer on large-scale molecular data[J]. Advances in neural information processing systems, 2020, 33: 12559-12571.
> > >
> > > [2] Understanding the limitations of deep models for molecular property prediction: Insights and solutions. NeurIPS 2023.
> > >
> > > [3]   Enhancing activity prediction models in drug discovery with the ability to understand human language. ICML 2023.

---

> > > > ### Author Response · Authors · 2024-11-19
> > > > **New results**
> > > >
> > > > Thank you for your suggestions. We have updated Appendix D.3 with Table 7, providing new experiments comparing fully fine-tuned UniMol, InfoAlign, and various representations. Here, we update the response in "W2, Q4: Feature Concatenation." The new results are presented in the table below:
> > > >
> > > > | Fine-tuning Method                         | ChEMBL2K  | Broad6K  | ToxCast   | Biogen3K  |
> > > > |--------------------------------|-----------|----------|-----------|-----------|
> > > > | Representation from UniMol          | 76.8±0.4  | 65.4±0.1 | 64.6±0.2  | 55.8±2.8  |
> > > > | Representation from UniMol and Other Features | 77.5±0.1  | 66.4±0.5 | NA        | NA        |
> > > > | Fully-tuned UniMol        | 78.9±0.2  | 65.1±1.0 | 71.3±0.6  | 43.6±0.3  |
> > > > | Representation from InfoAlign      | 81.3±0.6  | 70.0±0.1 | 66.4±1.1  | 49.4±0.2  |
> > > > | Fully-tuned InfoAlign      | 80.1±0.9  | 69.2±0.7 | 72.0±0.5  | 42.8±1.1  |
> > > >
> > > > From the table, we observe that fully fine-tuning benefits both UniMol and InfoAlign on ToxCast and Biogen3K datasets. While fully fine-tuning improves the representations of UniMol on ChEMBL, InfoAlign representations, with only MLP decoder tuning, achieves the best performance. On Broad6K, fully fine-tuning is less effective for both InfoAlign and UniMol compared to tuning only the MLP. These results suggest that, if resources allow, fully fine-tuning should be preferred for better performance, especially for UniMol, which requires more time and resources due to the use of 3D molecular structures. If resources are limited, InfoAlign's representation provides a strong alternative without full fine-tuning.
> > > >
> > > > We hope these new results help address the reviewer’s question:
> > > >
> > > > > "If you have collected so much multimodal data and constructed complex pre-training tasks, but they still cannot surpass the work of others that use single-modality pretraining plus full fine-tuning, then why go through all this trouble to collect this data? I only need to use single-modality pre-training plus full fine-tuning, which is more in line with Occam's Razor."
> > > >
> > > > The new results demonstrate the value of molecular pretraining with multimodal data from cellular responses, both for representation learning and full model fine-tuning. Additionally, we believe UniMol also highlights the value of collecting other multimodal data, such as molecular 3D structures, for improving pretraining.
> > > >
> > > > > "Moreover, single-molecule data is easier to collect and more abundant than biological data, making it easier for people to scale up their models. So, if your method cannot outperform an approach that only uses molecular pre-training plus full fine-tuning, why would anyone choose to use your method, which lacks both scalability and performance?"
> > > >
> > > > As previously mentioned, the new results highlight the value of cellular response data. We also observe that InfoAlign is more efficient than UniMol during fully fine-tuning, as UniMol requires computing 3D molecular structures based on RDKit to obtain atomic coordinates. For example, on the ToxCast dataset, UniMol takes around 46-50 seconds per epoch, while InfoAlign only requires 2-3 seconds.
> > > >
> > > > > The provided references [2, 3] only demonstrate that fingerprints can indeed outperform simple GNNs such as GCN and GAT on some tasks, but they do not prove that fingerprints possess the capability to match pretrain-GNNs. Moreover, in terms of the magnitude of improvement, pretrain-GNNs can bring about a very significant enhancement [1], which is larger than the performance gap shown in [2]. Therefore, this does not constitute a valid reason for not comparing fully finetuned models.
> > > >
> > > > Thank you for your insightful observations. We have provided new results comparing fully fine-tuned models. In our previous response, our goal was to show that fingerprints can serve as a good baseline. We believe our statement, "Morgan fingerprints do not challenge the existence or progress of molecular pretraining," aligns with your observations.
> > > >
> > > > We appreciate your constructive and insightful feedback. We hope these new results address your concern regarding the comparison of fully fine-tuned models. We are happy to address any remaining concerns.

---

> > > > > ### Comment · Reviewer_Sxqu · 2024-11-20
> > > > >
> > > > > - The conformation can be directly computed from the SMILES, and in molecular representation learning, we do not consider this as information from another modality. Moreover, Uni-mol only requires atomic coordinates and does not need any additional information.
> > > > > - The application scenario for molecular property prediction is to provide preliminary screening or priority ranking for downstream  experiments in the wet lab, which can take days or even months to complete, especially when these properties are related to biology. No one cares whether your model takes one second or one minute to make predictions, as this is negligible compared to the costs associated with the subsequent wet experiments in the wet lab.

---

> ### Comment · Reviewer_Sxqu · 2024-11-19
>
> - The paper focuses on molecular representations, but if such a representation cannot outperform the features obtained from full fine-tuning, then what is the significance of such a kind of representation? If you have collected so much multimodal data and constructed complex pre-training tasks, but they still cannot surpass the work of others that use single-modality pretraining plus full fine-tuning, then why go through all this trouble to collect this data? I only need to use single-modality pre-training plus full fine-tuning, which is more in line with Occam's Razor.
> - Moreover, single-molecule data is easier to collect and more abundant than biological data, making it easier for people to scale up their models. So, if your method cannot defeat an approach that only uses molecular pre-training plus full fine-tuning, why would anyone choose to use your method, which lacks both scalability and performance?

---

> ### Comment · Reviewer_Sxqu · 2024-11-20
>
> - The supplementary experimental results in the authors' rebuttal have demonstrated that **full-finetuning can significantly enhance the performance of the baselines**. However, the paper **does not present the performance of fully finetuned baselines**. The presented performance of pretrained-GNNs is **substantially underestimated**. This means the paper lacks sufficient evidence to show that the proposed methods, which require more complex multi-modal data construction, demonstrate a clear advantage over simple single-molecule pretraining.
> - Additionally, the supplementary experimental results indicate that after full finetuning, the proposed method only provides a **marginal performance improvement** on 3/4 of the datasets used in the paper compared to Uni-Mol, not to mention that Uni-Mol is not the most outstanding among all pretrained-GNNs. According to Occam's razor, which is agreed upon by both reviewers and authors, it is difficult to justify the necessity of the proposed complex multi-modal pretraining, especially when the collection of biological experimental data is much more challenging than that of single-molecule data.
> - Even if the authors were to update the performance of all pretrained-GNNs, **nearly half of the experimental results would need to be modified**, and consequently, the experiment analysis would also require extensive revision. If the authors were to undertake such a large-scale rewrite of the paper, the review provided by the reviewer would no longer be applicable to this article, necessitating new reviewers and a new round of peer review, which would violate the ICLR submission process.
>
> For the above three reasons, I will  downgrade my score to reject.

---

> ### Author Response · Authors · 2024-11-20
> **Kindly request a closer review of the contribution regarding representation learning instead of full-tuning**
>
> Thank you for your thoughtful comments and questions. To clarify, we would like to briefly reiterate the key points previously discussed:
>
> - The primary focus of the paper is representation learning, with an emphasis on the quality of molecular representations.
> - We have ensured fair evaluation by using consistent fine-tuning methods with MLP task predictors.
>
> We are happy to continue the discussion on full fine-tuning, at the same time, we wish to maintain the focus on representation learning. To summarize the distinction:
>
> - Full fine-tuning optimizes the encoder parameters for downstream tasks, which may lead to biased evaluation due to the use of stronger encoders.
> - In contrast, representation learning aims to achieve high-quality representations even with simpler encoders, such as the GIN used in this paper.
>
> **Fully fine-tuning and representations can be studied separately**. Advancements in both areas can contribute to the field. Real-world virtual screening is more complex than relying on a single representation or a fully fine-tuned model. Ensemble methods that combine both fine-tuning and representations offer advantages in both areas. **However, understanding the rationale behind each component is essential. This paper focuses on representation learning with consistent, fair evaluations. In this context, InfoAlign has shown strong performance through extensive experiments in Table 1/2 and Figure 3, including 27 baselines.**
>
> As requested by the reviewer, we provide new results **extending** beyond the main contribution of representation learning, focusing on fully fine-tuning. The results show that the simpler GIN encoder in InfoAlign outperforms the Transformer used in UniMol, even in fine-tuning scenarios, highlighting the superiority of InfoAlign as a pretraining method.

---

> > ### Author Response · Authors · 2024-11-20
> > **Regarding the reviewer’s new comments**
> >
> > > The conformation can be directly computed from the SMILES, and in molecular representation learning, we do not consider this as information from another modality. Moreover, Uni-mol only requires atomic coordinates and does not need any additional information.
> >
> > While it is possible to generate a conformation from a SMILES string using tools such as RDKit, this process introduces additional assumptions and approximations. SMILES itself does not contain explicit 3D spatial information, so generating a conformation requires computational steps like energy minimization or force-field modeling, which depend on assumptions about molecular geometry.
> >
> > > The application scenario for molecular property prediction is to provide preliminary screening or priority ranking for downstream experiments in the wet lab, which can take days or even months to complete, especially when these properties are related to biology. No one cares whether your model takes one second or one minute to make predictions, as this is negligible compared to the costs associated with the subsequent wet experiments in the wet lab.
> >
> > In real-world applications with millions of virtual screening candidates, inference time becomes critical. Even a 1-second inference time results in 11 days for one million candidates. If more accurate 3D information is required, methods like DFT calculations or experimental techniques (e.g., X-ray crystallography) can take much longer to generate accurate atom coordinates, potentially making the inference time unacceptable.
> >
> > > The supplementary experimental results in the authors' rebuttal have demonstrated that full-finetuning can significantly enhance the performance of the baselines. However, the paper does not present the performance of fully finetuned baselines. The presented performance of pretrained-GNNs is substantially underestimated.
> >
> > Based on results from ChEMBL2K and Broad6K, fully fine-tuning does not always lead to improvement.
> >
> > Real-world deployment of accurate virtual screening is complex, with ensemble methods offering advantages in both fine-tuning and representations. However, understanding each component's rationale is crucial, and this paper focuses on the representation learning aspect.
> >
> > > This means the paper lacks sufficient evidence to show that the proposed methods, which require more complex multi-modal data construction, demonstrate a clear advantage over simple single-molecule pretraining.
> >
> > **InfoAlign is a representation learning method, shown to outperform 27 different representations from various methods, as demonstrated in the original paper**
> >
> > As an extension, we have added new experiments comparing the encoders from InfoAlign with UniMol in fully-tuned scenarios, where the InfoAlign model also shows consistently improved performance.
> >
> > Regarding data construction complexity, the contribution of this paper is not about generating new data points for cell morphology or gene expression. We have curated existing data for pretraining, which does not involve biological experiments.
> >
> > Regarding algorithmic complexity, the random walk approach is efficiently implemented using a sparse graph and introduces minimal computational complexity in both time ($\mathcal{O}(k)$) and space ($\mathcal{O}(M)$), where $k$ is the average degree and $M$ is the number of edges.
> >
> > > Additionally, the supplementary experimental ... pretrained-GNNs.
> >
> > **The new observations regarding full fine-tuning should not alter our current conclusions on representation learning.**
> >
> > We apologize for any confusion. We chose UniMol for comparison with InfoAlign as it seems to be the reviewer’s preferred baseline. If the reviewer has other suggestions, we would be happy to discuss them further. However, we believe such discussions do no affect the main focus of the paper on representation learning.
> >
> > > According to Occam's razor ... challenging than that of single-molecule data.
> >
> > **For evaluating representation learning**, we agree with the principle of Occam's razor, as full fine-tuning may lead to biased evaluation due to stronger encoders. In contrast, representation learning aims to achieve high-quality representations even with simpler encoders.
> >
> > Regarding complexity, both data and algorithmic complexity are not the concerns, as discussed earlier.
> >
> > > Even if the authors were to update ... ICLR submission process.
> >
> > We appreciate the reviewer’s suggestion and are happy to discuss InfoAlign’s potential in fully-tuned settings, as it could enrich the conversation. However, we respectfully note that such a discussion should not shift the main focus of the paper, which is on representation learning, including but not limited to the results.
> >
> > Exploring InfoAlign in fully-tuned settings should be considered an extension of the current work, and we believe such discussions would require only minor modifications to the paper.

---

> > > ### Comment · Reviewer_Sxqu · 2024-11-21
> > >
> > > > Regarding data construction complexity, the contribution of this paper is not about generating new data points for cell morphology or gene expression. We have curated existing data for pretraining, which does not involve biological experiments.
> > >
> > > Existing biological data is the result of extensive laboratory work conducted by numerous individuals in the fields of biology and chemistry. In contrast, the collection of single-molecule data often does not rely on traditional wet-lab experiments and can be achieved through computational simulations alone. The generation of biological data is not a simple task; if you believe that these data are easily collected, you are merely benefiting from the work of countless preceding researchers. Your viewpoint appears to dismiss the efforts of biochemistry professionals and is fraught with arrogance and a lack of understanding. As a fellow researcher, how could you make such a statement?

---

> > > > ### Author Response · Authors · 2024-11-21
> > > > **Experiments**
> > > >
> > > > We thank the reviewer for their insightful comments and apologize for any miscommunication in our initial response. We aim to address the limitations of early-stage experiments and high-throughput data generation for large-scale libraries. As the reviewer rightly pointed out, generating data from assays like L1000 and Cell Painting is time-intensive, often requiring months to procure compounds, prepare cell lines, and execute screens.
> > > >
> > > > One alternative, as suggested, is machine learning (ML)-based virtual screening, which often utilizes chemical structures as inputs. These models leverage the chemical similarity principle (i.e., similar compounds exhibit similar activity) to make predictions. However, datasets used in such models frequently suffer from analogue bias and other challenges, such as incomplete stereochemical information in SMILES representations. As a result, these models often struggle to generalize beyond the chemical space covered in the training data.
> > > >
> > > > This limitation brings us back to the necessity of generating biological data to expand the applicability domain of ML models. While it is infeasible to experimentally screen the vast chemical space (~10^40 molecules), incorporating existing experimental data into representation learning provides a practical path forward. Our work focuses on integrating experimental data to enhance molecular representations, enabling the generation of descriptors enriched with both biological and chemical information. By doing so, we aim to reduce the need for extensive experimental campaigns in the future while facilitating rapid and scalable virtual screens, while having models that outperform previously published models.

---

> ### Author Response · Authors · 2024-11-21
>
> Dear reviewer Sxqu,
>
> I am a senior author on the paper. I greatly appreciate your taking the time to discuss our work.
>
> The concerns that my lab member’s “viewpoint appears to dismiss the efforts of biochemistry professionals and is fraught with arrogance and a lack of understanding” are misplaced - our laboratory has actually led the experimental work for a large proportion of the publicly available image data of this type, so we are all aware of the challenges and value of lab work. I hope this is a simple mis-reading of my lab member’s response as I don’t see anything in our note to warrant this response.
>
> Now back to the debate:
>
> I believe your major concern this: if fully tuned models are doing better than our learned representations, then our learned representations are not terribly useful, and are even less useful if the representations are cumbersome to generate, i.e, they need data collected from wet lab experiments.
>
> If that is indeed your primary concern, I think that's a valid one and I am happy to clarify our stance further, below.
>
> ---
>
> **Concern 1: If fully tuned models perform better than our learned representations, then our learned representations are not terribly useful**
>
>
> 1. Representation learning supports a broader range of use cases. Given the vast number of unlabeled molecules, pre-trained representations can be efficiently stored and applied to various downstream tasks, such as visualization in virtual screening analysis.
>
> 2. In fully fine-tuned scenarios, representations from pre-trained models can still enhance performance. For example, on ChEMBL2K and Broad6K, InfoAlign’s representations (81.3±0.6 and 70.0±0.1) outperform fully-tuned UniMol (78.9±0.2 and 65.1±1.0) by 3.0% and 7.7%, respectively. In the NLP community [1], research shows that the usefulness of each paradigm depends on the specific downstream task. Therefore, both paradigms offer distinct advantages and can be combined. Representation learning does not conflict with full fine-tuning.
>
> **Concern 2: The representations are even less useful if the they are cumbersome to generate, i.e, they need data collected from wet lab experiments.**
>
>
> I think this is a fair criticism -- it's definitely a non-trivial overhead if one's approach needs data collected from wet lab experiments.
>
> However,
>
> 1. The data we used is already publicly available; we simply tapped into existing resources.
> 2. Profiling assays like Cell Painting are now being routinely run by several academic labs and pharma companies, sometimes for quite large subsets of their compound libraries. Thus, in some contexts this data is freely available to their scientists for compounds of interest. What an amazing opportunity researchers have to tap into that, using InfoAlign-like methods!
> 3. InfoAlign can produce embeddings for a molecule even if we don't have a Cell Painting/gene expression profile for the molecule; hopefully this bit was already clear.
>
> Let us know what you think, and thank you for remaining engaged!
>
> ---
>
> [1] To tune or not to tune? adapting pretrained representations to diverse tasks. ACL. 2019.

---

### Official Review · Reviewer_VNuN · 2024-10-30

**Soundness:** 3
**Presentation:** 3
**Contribution:** 2
**Rating:** 6
**Confidence:** 3

**Summary:**

Thank you for a really interesting read, I agree that the most important next step in molecular representation is combining different modalities and current methods leave much to be desired, and I really liked the novel way shown here to combine data in a constricted way. It is well mathematically motivated and the work done clearly extensive.

Summary:
* The authors present a new multi-domain method to learn the representation between drugs, genes and cells. In contrast to typical contrastive training setups the InfoAlign method removes the redundant information through a bottleneck using a well formulated mutual information method.
* They show a method to create walks over the cellular context  graph representing the interactions between the modalities, and use this to populate the compute graph for their representation learning.
* This representation framework uses Morgan fingerprints as molecular node features, CellProfiler features for the cell node features, and L1000 features for the gene node features, and the connections are based on chemical perturbations and cosine similarities.
* The authors show results across a range of benchmarks, demonstrating good performance, achieving top ranking scores in all but 2 / 15 of the key datasets/criteria evaluated in Tables 1 and 2.

**Strengths:**

* The novel construction of the problem mitigates a significant issue with current implementations using a standard contrastive learning approach, by combining the modalities into a single method there is no decoupling between the different representations, and the MI approach to filter the data seems an effective way to reduce the data required to train over, potentially increasing data quality and improving time to train.
* The visual presentation is good, with tables are figures clearly designed to convey meaning.
* The breadth of models evaluated against gives a really clear status against both several standard approaches one might take, and also against SOTA prior work. This comparison is really nice to see.

**Weaknesses:**

* Model clarity - I found it hard reading this paper to extract exactly how the model was constructed to run these experiments. The method is clearly novel, and an interesting approach to the multi-domain problem, but after being strongly theoretically motivated the method / implementation details, or even a description of model size was lacking. If these details could be expanded on it would help place the model / method in the appropriate context.
* Impact of the MI bottleneck - The mutual information bottleneck is well theoretically motivated, however I didn’t feel the power of the bottleneck was evaluated in the paper? A different approach would be just using all the data and relying on data size / model size to outweigh data quality. I like the idea with MI, but would have liked to see an ablation with this feature included / not as that would help me evaluate the importance of the bottleneck vs the joined training setup?
* Impact of context graph - Similarly the impact of the context graph and different ways of including the context felt under-explored. Fig 4(b) I think shows that the random walk length had little impact, but how this is impacted by composition of the random walk path (are the walks just reconstructing the most likely combinations that would form a triplet anyway?) and what happens if instead of walking the relevant combinations are just grouped together. This feels like an important baseline.
* The presentation of the research questions felt more like a report than a paper, I would have liked to see some more motivation / explanation of each rather than assuming knowledge on the readers part.

**Questions:**

Questions / suggestions:
* I would ask for some more clarity on the results presented in table 1, specifically for the Broad6k results. Many columns have a +/- 0 error on a reoccurring 3.1 result. I suspect this means that there are a few tasks that are always identified with a high AUC, but given the lack of discussion of these results it’s hard to interpret. It might be worth using a different set of thresholds for the Broad6K to get more resolution?
* It would also help in section 6 for each of the research questions to have a couple of sentences explaining what each question is and why we are asking them, otherwise the results feel disjointed to the uninitiated reader and hard to connect.
* I found the explanation of the exact model architecture / parameters used to be lacking, and while pointed in the main text to the appendix I did not find enough information there on the size of/ structure of the MLPs / training regime to feel like the result could be replicated. This is of particular importance in Table 1 where comparing to other methods I can only guess at things like parameter efficiency etc.

* In general I feel like the weighting of the paper is very theoretical, no bad thing I really like the inclusion of the mutual information, but I wonder if more details could be moved to the appendix to free up more space for the experimental method, the model construction, and discussion of the results. In the current format of the paper I find a really interesting set of ideas, that I find very hard to understand how to weight the importance of / a method that I could follow to replicate certain components.


Specific formatting suggestions:
* Fig. 3. I don’t really understand what this graph is conveying? The top bar shows the proportion of single representation tasks, but the bottom splits along the model types? I either need more detailed explanation, or maybe consider a better type of graph to convert this information?
* Table 3. (right) (This should be a separate figure - I understand it might make formatting harder but it is hard to reference) I personally don’t like these KDE type plots as they imply smooth functions from what is usually limited data, and make it almost impossible to draw quantified conclusions from, I would be much happier with a histogram if this is important information to convey. Additionally x-axis labels.
* Fig 4 (a) - this plot is almost unreadable with the overlapping lines / y-scale, and the meaning I think is being conveyed with the LR spans such a number of magnitudes in range that I would rather see both more granularity and perhaps these results presented in a table?
* Fig 4 (b) - I can guess what the plot means but without a key / description detailing elements like the error band (? 1 std dev I assume, based on what variation is unclear though), the points are connected, but given the discrete data this is misleading, and the huge discrepancy at length 8 suggests to me either the error band is under-estimating the variance, or there are properties of a random walk length 8 that are not discussed. Again this is a really interesting result as it's looking at the way that different parts of the compute graph contribute to the final result, but I'm left asking more questions with this figure than it answers.
    * While I thin this statement on line 521 is correct, “we observe in Figure 4b that downstream performance on ChEMBL2K is relatively robust across a wide range of walk lengths.” The inclusion of this plot raises questions that are not answered.
* On a similar point to Fig 4b, I would be very interested to see a description / plot of what the typical construction of the random walk graphs contain.



Thank you again for a really interesting read, I like the approach with InfoAlign, and appreciate the large quantity of effort put into this work.
I'd request a slight look again at the weighting of different sections in this paper, as I found it slightly unbalanced, and think as a reader I would find benefit from more detail in the method so I could reproduce if I wanted, and more detail in the discussion of the results to understand how to place the results in better context. i.e. which parts of the method are the most important.

---

> ### Author Response · Authors · 2024-11-18
> **Author Response [1/3]**
>
> We are pleased that you found our work interesting and appreciate your thoughtful comments and suggestions for improving the paper. We have revised the paper accordingly and provided a point-by-point response. We believe after revision, the current structure of the paper is well-balanced. If any concerns remain, we welcome further discussion to address them. Thank you again for your constructive feedback.
>
> ## W1, Q3: Model clarity
>
> The model setup is described in Line 307-312: we use a Graph Isomorphism Network (GIN) as the encoder and an MLP as the decoder. **All details about model configurations, including layers and hidden dimensions, are provided in the supplementary code, along with a pretrained checkpoint** for easy reproducibility. Below, we report the hyperparameters from the code:
> | Parameter                                                       | Value        |
> |-----------------------------------------------------------------|--------------|
> | hidden dimension                                                | 300          |
> | normalization layer                                             | batch norm   |
> | number of layers                                                | 5            |
> | node-to-graph readout                                           | sum          |
> | $\beta$ (in Eq. (3) second term)                                | 1.0e-09      |
> | $L$ (Walk length)                                               | 4            |
>
> The MLP consists of three layers: an input dimension of 300, a hidden layer with a dimension of 4 × 300, and an output layer corresponding to the feature/task dimension. This MLP architecture is generally applied on all representations.
>
> We apologize for the oversight in referencing the code. We have updated Section 5 and Appendix B.3 with the relevant model information. In our attempt to stay anonymous, the complete code is available in the supplementary materials and we will include the code link in the camera-ready version.
>
> ## W2: Impact of the MI bottleneck
>
> This work is motivated by the principle of the information bottleneck, which leads to the optimization targets in Eq. (3) for molecular representation learning and pretraining.
>
> ### The effectiveness of the bottleneck was evaluated by comparing InfoAlign with multi-modal contrastive alignment approaches.
>
> For joint use of data in pretraining, Figure 1 and Lines 85–102 show that the proposed architecture is guided by the information bottleneck principle, which includes a single encoder with multiple decoders. Without this principle, jointly using two data types would be contrastive learning approaches, such as InfoCORE and CLOOME (Figure 1(a)). Comparing these baselines in Tables 1 and 2 thus allows us to assess the impact of the information bottleneck. We also note that methods for jointly integrating molecular, cell morphology, and gene expression data in contrastive learning remain underdeveloped.
>
> ### We focus on applying the information bottleneck principle during the pretraining.
>
> Joint use of data in downstream tasks, such as ToxCast and Biogen3K (Table 2), is often not feasible due to the high cost of obtaining cell morphology and gene expression data from biological experiments, compared to extracting molecular structure features. This is why we apply the information bottleneck during pretraining, rather than in downstream tasks. We agree that training models with MI constraints in downstream tasks, instead of using joint features, could be a promising approach. However, this is beyond the scope of our current focus on pretraining and may be explored in future work.

---

> > ### Author Response · Authors · 2024-11-18
> > **Author Response [2/3]**
> >
> > ## W3, Q9: Impact of context graph (construction of the random walk graphs)
> >
> > Thanks for your insightful comments. We conducted new experiments and found that random walks provide diverse neighbors, improving the pretraining performance. We have included the new results in Appendix D.5 and added a reference to the appendix in Section 6.3.2 of the main text.
> >
> > ### Random walk produces diverse neighbors
> >
> > We cached the random walk results for 100 epochs and studied the number of unique nodes at varying walk lengths. In this table, we report the mean and standard deviation (STD) of unique nodes for all pre-training molecules at each walk length.
> >
> > | Walk Length | Unique Nodes (mean±std)             |
> > |-------------|-----------------------------|
> > | 2           | 8.39±23.51            |
> > | 3           | 13.31±41.02           |
> > | 4           | 18.41±59.83           |
> > | 5           | 23.20±77.74           |
> > | 6           | 28.12±96.31           |
> > | 8           | 37.30±131.15          |
> > | 10          | 46.03±164.46          |
> > | 12          | 54.35±196.25          |
> >
> > (Note that the minimum number of unique nodes is 1 for isolated nodes)
> >
> > If the composition of the random walk path were fixed, the number of unique nodes would be close to the walk length. However, we observed that the number of unique nodes is larger and varies, suggesting that diverse nodes are included in the random walk paths.
> >
> > We further explored the Jaccard similarity of neighborhoods extracted for the same molecule under varying walk paths, averaging similarity scores across all pretraining molecules. The pairwise similarities for different walk lengths are shown in the table below. We observe that similarity decreases as the difference in walk lengths increases, but remains above 90%. This may explain the stable performance of InfoAlign in Figure 5(b) (original Figure 4 (b)). These results suggest that even with a walk length of 2, diverse neighbors can be obtained, likely due to the presence of high-degree nodes in the context graph.
> >
> > | $L$ | 2 | 3 | 4 | 5 | 6 | 8 | 10 | 12 |
> > |--------|--------|--------|--------|--------|--------|--------|--------|--------|
> > | 2 | 100.0 | 92.5 | 92.0 | 91.8 | 91.7 | 91.5 | 91.4 | 91.3 |
> > | 3 | 92.5 | 100.0 | 93.2 | 93.0 | 92.8 | 92.6 | 92.5 | 92.3 |
> > | 4 | 92.0 | 93.2 | 100.0 | 93.4 | 93.3 | 93.1 | 92.9 | 92.8 |
> > | 5 | 91.8 | 93.0 | 93.4 | 100.0 | 93.4 | 93.3 | 93.2 | 93.1 |
> > | 6 | 91.7 | 92.8 | 93.3 | 93.4 | 100.0 | 93.5 | 93.4 | 93.3 |
> > | 8 | 91.5 | 92.6 | 93.1 | 93.3 | 93.5 | 100.0 | 93.6 | 93.6 |
> > | 10 | 91.4 | 92.5 | 92.9 | 93.2 | 93.4 | 93.6 | 100.0 | 93.7 |
> > | 12 | 91.3 | 92.3 | 92.8 | 93.1 | 93.3 | 93.6 | 93.7 | 100.0 |
> >
> > ### Fixed neighbors underperform random walk-sampled neighbors.
> >
> > Regarding the ablation study on the importance of diverse neighbors with random walk sampling, we conducted additional experiments. During pretraining, we randomly selected 4 direct neighbors and fixed them, instead of performing a random walk with a walk length of 4. The results, presented in the table below, highlight the importance of diverse neighborhoods extracted by the random walk for improved performance.
> >
> > |              | ChEMBL2K       | Broad6K        | ToxCast        | Biogen3K       |
> > |---------------------|----------------|----------------|----------------|----------------|
> > | Random Walk      | 81.33±0.62     | 69.95±0.09     | 66.36±1.05     | 49.42±0.18     |
> > | Fixed Neighbors   | 77.47±0.38     | 66.75±0.13     | 65.43±0.76     | 50.08±0.30     |
> >
> > In summary, random walks improve performance by sampling diverse neighbors. We appreciate your comment and welcome further discussion.
> >
> > ## W4 and Q2: Presentation of research questions
> >
> > Thanks for your comments. We have updated the main text with an explanation of the research questions at the beginning of Section 6, as well as at the start of Sections 6.1 and 6.2. These updates are provided below for your reference.
> >
> > Section 6: "We demonstrate the effectiveness of InfoAlign's representation in (1) molecular property prediction, (2) molecule-morphology matching, and (3) analyze the performance of InfoAlign. These lead to three research questions (RQs)."
> >
> > Section 6.1: "Better molecular representations should improve prediction performance. We train MLPs on different representations to predict molecular properties in both classification and regression tasks.."
> >
> > Section 6.2: "Molecular representations are aligned with cell morphology. The zero-shot matching performance of a queried molecule to cell morphology features evaluates the alignment between the two modalities."

---

> ### Author Response · Authors · 2024-11-18
> **Author Response [3/3]**
>
> ## Q1: Results in table 1
>
> Thank you for the question. There are 32 tasks for Broad6K, and the value 3.1 means that one task (3.125% = 1/32 * 100) is frequently predicted with high AUC (>90%). The task ID is 274_752. Using metadata from Broad6K, this task involves an MLPCN LGR2 assay, which aims to identify compounds targeting the LGR2 GPCR protein. This assay is an antagonist of the LGR2 target which will then compromise survival for pests like ticks or mosquitoes.
>
> Regarding resolution, we observe that 3.1 occurs frequently with a threshold >85%, and the current resolution is sufficient to reflect the high AUC for this specific task.
>
> ## Q4: Weighting of the paper
>
> Thanks for your suggestions. Currently, the main text consists of 2.8 pages for background (Introduction, Related Work, problem definition), 3 pages for methods (1.7 pages for methods, 0.5 pages for theoretical outcomes, and 0.8 pages for model implementation), and 4.2 pages for experimental results.
>
> For the theory, Section 4.3 presents the key theoretical outcomes in less than half a page, with further details in Appendix B (formerly Appendix A).
>
> We have added more details on model construction, with references to the appendix for additional information. The code provides more clarity than text descriptions alone. We have included the code with checkpoints for easy reproducibility in the supplementary materials and will add the link in the camera-ready version.
>
> We have updated Figures 3, 4, 5, and Table 4 for better clarity. We also included random walk analysis in Appendix D.5 and relocated some related content to the appendix to accommodate page limitations.
>
> The current page allocation for background, methods, and experiments is 2.8:3:4.2, which we believe is well-balanced, with a emphasis on experimental results. We appreciate your comment and welcome any further discussion on adjusting the paper’s content for better presentation.
>
> ## Q5: Figure 3
>
> We have updated the caption for improved clarity. Figure 3 compares the relative performance of two groups of models: (1) representations using single-modal information (Single Rep.) and (2) molecular representations from multi-modal alignment methods (Aligned Rep.). The top bar compares models using single-modal information from the best baselines in Molecular Structure, Cell Morphology, and Gene Expression. The bottom bar compares three models that use multimodal information.
>
> ## Q6: Table 3 right part
>
> We have separated the figure and table, resulting in the current Table 3 and Figure 4. We also updated the distribution figure to be a histogram and added x-axis labels. We appreciate your suggestion, and with these changes, we believe Figure 4 now better illustrates the observations in Section 6.2.2.
>
> ## Q7: Figure 4 (a)
>
> In Figure 4(a) (now Figure 5 (a)), $\beta$ refers to the hyperparameter from the second term in Eq (3), not the learning rate. The figure shows how pretraining losses change with varying regularization strengths, which may not be as clearly represented in tables.
>
> From the figure, we observe that pretraining loss may serve as an indicator for selecting $\beta$. Specifically, for $\beta = 1e-9$ and $\beta = 1e-12$, lower pre-training losses correspond to better downstream performance.
>
> We have updated the figure to highlight the pretraining losses for $\beta = 1e-9$ and $\beta = 1e-12$ in the revision to address your concerns.
>
> ## Q8: Figure 4 (a): walk length
>
> Thanks for your comments. Each result is based on ten runs, with the points representing the mean value and error bars showing one standard deviation. We have updated the figure and caption for clearer representation.
>
> Figure 4(b) (now Figure 5(b)) is primarily intended to support our claim of robust performance across different hyperparameter choices for $L$, as confirmed by the reviewers. As discussed in our response to W3/Q9, we do not observe any properties indicating that length 8 is an outlier. While the error bar may not fully capture the variance, we believe the variation at length 8 does not affect the comparison and observation with the best baseline.
>
> Further discussions on the other questions about the figure are provided in the responses to W3/Q9.

---

> > ### Comment · Reviewer_VNuN · 2024-11-25
> >
> > I thank the authors for their considered response to the comments and questions I provided.
> >
> > I believe the changes made make for a more readable paper, and significantly improve the reproducibility for which I am very grateful.
> > The detailed response puts the numerical results in better context with suitable error bars to understand the distributions better, and the authors also addressed several key issues with figures which made understanding the results more challenging, so on reflection I am very happy to raise my score to reflect this change.
> >
> > Thank you again for a very enjoyable read, and such good engagement in the review process.

---

> > > ### Author Response · Authors · 2024-11-25
> > > **Thank you for raising the score**
> > >
> > > Thank you for raising the score. We sincerely appreciate the reviewer’s thoughtful engagement and constructive feedback throughout the discussion. We are pleased that the reviewer’s concerns have been addressed and are grateful for the support of our work!

---

### Official Review · Reviewer_ALhV · 2024-11-01

**Soundness:** 3
**Presentation:** 4
**Contribution:** 3
**Rating:** 8
**Confidence:** 4

**Summary:**

The authors propose a method called InfoAlign for predicting molecular properties by integrating three different modalities: molecular structures, gene expression, and phenomics embeddings. To learn useful representations, they construct a weighted connected graph over cell morphology profiles, related molecules, and gene expression values. They train an encoder-decoder architecture where, for a molecule of interest, they encode its representation and decode both itself and all other nodes encountered during a random walk on a pre-specified graph. This approach results in mutual information maximization between the compound of interest $x_i$ and related entities discovered through the random walk on the pre-encoded graph. The authors test their method on a variety of chemical property prediction datasets, demonstrating that they outperform various baselines, including pre-trained Graph Neural Networks (GNNs), chemical language models, uni-modal models such as cell morphology or gene expression, and some multi-modal alignment models.

**Strengths:**

- The authors perform a comprehensive evaluation against baseline models across a variety of datasets.
- They convincingly demonstrate that including additional modalities improves performance, as evidenced by thorough evaluations and ablation studies.
- The incorporation of a graph is an interesting way to introduce prior knowledge into the learning representations for a particular molecule.

**Weaknesses:**

- The validity and robustness of the pre-specified graph are not thoroughly explored. It would be informative to assess how sensitive the method is to the quality of the graph. For example, one experiment could involve removing 50% of valid connections and replacing them with random pairs of nodes; another could involve using a completely random graph.
- The second gap identified by the authors is slightly misformulated: "They treat molecules as the sole connectors between gene expression and cell morphology, ignoring the potential for genetic perturbations."
    - Essentially, the authors are arguing that incorporating genetic perturbation data can further improve predictive capacity. However, there is no ablation study where this information is omitted to directly validate its impact on empirical performance.
- Regarding the ToxCast dataset, the authors report a performance of 0.72 ROC AUC using GROVER. Did the authors use a different partitioning of the dataset than previous works?
- The ablation loss is only regarding removal of the losses as far as I understand the data itself is still input into the training. Can the authors perform an ablation where a full data modality is not added as part of training?

**Questions:**

- Are the authors surprised by the relatively minor drop in ROC AUC values when omitting individual modalities?
- Some additional relevant literature that could enhance the discussion includes:
    [0] Cross-Modal Graph Contrastive Learning with Cellular Images
    [1] How Molecules Impact Cells: Unlocking Contrastive PhenoMolecular Retrieval (a work evaluating zero-shot classification)
    [2] Approximating Mutual Information of High-Dimensional Variables Using Learned Representations (proposes a scalable approach for approximating mutual information of high-dimensional objects)
- One of the conclusions from the work is emphasizing the importance of molecular features. Dot he authors have an explanation for why the absence of molecular features in the ablation results in a ToxCast AUC that overlaps the non-ablated model performance? Without seeing the results I would expect the performance in the absence of an ablated molecular feature reconstruction loss to be a lot worse.

---

> ### Author Response · Authors · 2024-11-18
> **Author Response**
>
> We sincerely thank the reviewer for their insightful suggestions. We provide point-by-point responses and have revised the text and appendix, highlighting changes in blue.
>
> ## W1: Random pre-specified graph
>
> We conducted the requested experiments, and the results are shown in the table below. We find that randomly replacing edges significantly impacts prediction performance. The replaced edges lack the biological, chemical, and computational meaning of those used to construct the context graph in the paper.
>
> |             | ChEMBL2K     | Broad6K      | ToxCast      | Biogen3K     |
> |--------------------|--------------|--------------|--------------|--------------|
> | No Random Edges (InfoAlign)         | 81.33±0.62   | 69.95±0.09   | 66.36±1.05   | 49.42±0.18   |
> | 50% Random Edges    | 77.48±0.71   | 62.52±0.16   | 63.76±0.32   | 71.53±2.51   |
> | 100% Random Edges   | 76.21±0.71   | 62.97±0.08   | 64.6±0.39    | 75.72±2.25   |
>
> ## W2: Genetic perturbation data
>
> In Table 4, we observe a performance drop when excluding gene expression data in loss functions.
>
> We also conducted new ablation studies, presented in Appendix D.4, with results shown in the table below. Removing all nodes related to genetic perturbation data from the context graph further decreases the performance, confirming the importance of genetic perturbation data.
>
> |                      | ChEMBL2K     | Broad6K      | ToxCast      | Biogen3K     |
> |-----------------------------|--------------|--------------|--------------|--------------|
> | w/o genetic perturbation data      | 77.97±0.33   | 67.1±0.17    | 64.93±0.96   | 51.57±0.46   |
> | InfoAlign                   | 81.33±0.62   | 69.95±0.09   | 66.36±1.05   | 49.42±0.18   |
>
> ## W3: GROVER performance on ToxCast
>
> The GROVER performance on ToxCast is reported as 53.1. We follow the standard splitting from Open Graph Benchmarking [1], which differs from the splitting used in the original GROVER paper [2]. Our results are consistent with those reported in [3].
>
> ## W4: Ablation studies on removing data
>
> We have conducted the requested ablation studies and have clarified them in Appendix D.4. The results are also shown in the table below. The first two rows display the removal of cell morphology or gene expression-related nodes from the context graph. We observe a further performance drop when these data are removed.
>
> |                      | ChEMBL2K     | Broad6K      | ToxCast      | Biogen3K     |
> |-----------------------------|--------------|--------------|--------------|--------------|
> | w/o cell-related nodes      | 79.57±0.58   | 68.41±0.31   | 65.11±0.82   | 51.21±0.17   |
> | w/o gene-related nodes      | 77.97±0.33   | 67.1±0.17    | 64.93±0.96   | 51.57±0.46   |
> | w/o cell-related loss       | 80.7±0.6     | 68.6±0.1     | 65.5±1.1     | 51.7±1.1     |
> | w/o gene-related loss       | 78.3±0.5     | 68.6±0.2     | 64.7±1.0     | 50.3±0.5     |
> | InfoAlign                   | 81.33±0.62   | 69.95±0.09   | 66.36±1.05   | 49.42±0.18   |
>
> ## Q1, Q4: Performance drop in ablation studies
>
> ### The performance of InfoAlign relies on the GNN encoder and different decoders with optimization targets.
>
> As shown in Tables 1/2 and Figure 3, GNN encoders can extract meaningful representations with proper loss designs. Ablation studies in Table 4 demonstrate that, even without one type of data, the remaining two types still form proper targets based on information bottleneck principles, supporting the pretraining of a good GNN encoder.
>
> ### We do not remove the GNN encoders, which continue to extract meaningful representations from molecular structures.
>
> In the ablation studies, the "absence of molecular features" refers to the removal of fingerprint vectors from the loss functions. In this case, cell morphology and gene expression can still optimize the GNN encoder for meaningful representation, as observed in previous work like InfoCORE [4].
>
> ## Q2 Relevant literature
>
> We found that they were all published or released this year, including the most recent NeurIPS 2024 [6]. We are happy to discuss them and have updated the related work and appendix accordingly.
>
> Lines 136-138: "CLOOME, MIGA [5], and MoCoP, and MolPhenix [6] contrast cellular images with molecules."
>
> Line 774: "Approximating the mutual information of high-dimensional variables is a challenging task [7]"
>
> ## Reference:
>
> [1] Open Graph Benchmark: Datasets for Machine Learning on Graphs. NeurIPS. 2020
>
> [2] Self-Supervised Graph Transformer on Large-Scale Molecular Data. NeurIPS 2020.
>
> [3] Evaluating Self-Supervised Learning for Molecular Graph Embeddings. NeurIPS 2023.
>
> [4] Removing Biases from Molecular Representations via Information Maximization. ICLR 2024.
>
> [5] Cross-Modal Graph Contrastive Learning with Cellular Images. Advanced Science 2024.
>
> [6] How Molecules Impact Cells: Unlocking Contrastive PhenoMolecular Retrieval. NeurIPS 2024.
>
> [7] Approximating mutual information of high-dimensional variables using learned representations.

---

> > ### Comment · Reviewer_ALhV · 2024-11-22
> >
> > > W1: Random pre-specified graph
> >
> > Thank you for a convincing rebuttal, demonstrating the utility of a valid graph. I found it interesting that graph is relatively resilient to random perturbation. My suspicion current entity graphs are incomplete and some of the conditions are spurious as they are observed under specific conditions.
> >
> > | Fine-tuning Method            | ChEMBL2K      | Broad6K       | ToxCast      | Biogen3K      |
> > |-------------------------------|---------------|---------------|--------------|---------------|
> > | Representation from UniMol   | 76.8±0.4      | 65.4±0.1      | 64.6±0.2     | 55.8±2.8      |
> > | 50% Random Edges             | 77.48±0.71    | 62.52±0.16    | 63.76±0.32   | 71.53±2.51    |
> >
> > Biogen task seems particularly sensitive to false edges.
> >
> > > W4: Ablation studies on removing data
> >
> > Model robustness to removal of individual data points is interesting as well. Do the authors think this is due to the shared information between different modalities?
> >
> > In my opinion this is an effective rebuttal and so I will raise my score. I've read the other reviewer's concerns regarding fine-tuning. I believe lack of fine-tuning is a fair criticism, but the authors do address it with an experiment demonstrating improvement over previous methods, albeit limited in scope. Relative strength of fingerprint methods is a known phenomenon in the field, however this does not invalidate the utility of alternative approaches.
> >
> > I think the paper presents a novel approach to an important problem and convincingly demonstrates improvements in quality of learned representations.

---

> > > ### Author Response · Authors · 2024-11-23
> > > **Thank you for your support and for raising the score**
> > >
> > > We appreciate the reviewer’s recognition of our rebuttal and insightful observations. The context graph, particularly edges based on computational criteria, could be improved in future work. For example, incorporating molecular properties, rather than just structural similarities, could address scenarios like activity cliffs, where molecules with similar features may differ in properties.
> > >
> > > Regarding model robustness, we aim to extract concise representations based on the information bottleneck principle, as illustrated in Figure 1. The loss function (Eq. 3) effectively extracts minimal sufficient information from two different modalities. In contrast to previous contrastive methods, which lack a term for minimal information and provide looser bounds on sufficient information, incorporating additional modalities could further improve the generalization of representations.
> > >
> > > We are encouraged that your concerns have been addressed. Thank you again for your support and for raising the score!

---

### Official Review · Reviewer_hxs4 · 2024-11-04

**Soundness:** 3
**Presentation:** 3
**Contribution:** 3
**Rating:** 6
**Confidence:** 3

**Summary:**

The paper introduces a novel approach called Information Alignment (InfoAlign) for learning molecular representations by integrating molecular structure, cell morphology, and gene expression data. The method leverages the information bottleneck principle to optimize a molecular graph encoder and multiple MLP decoders, aiming to achieve minimal yet sufficient molecular representations.

The authors demonstrate the effectiveness of InfoAlign through extensive experiments on molecular property prediction and zero-shot molecule-morphology matching, showing superior performance compared to 27 baseline methods across four datasets.

I find the paper of good quality in general. Besides, the problem it is tackling is meaningful but less explored. I suggest an acceptance to advocate this direction.

**Strengths:**

- The approach is well-motivated. Applying information bottleneck to this problem for learning minimal yet sufficient molecular representations seems a valid match of theory and real-world problem.
- The approach is comprehensively evaluated by comparisons across different methods and even paradigms. InfoAlign demonstrates improved accuracy over up to 27 baseline models across four datasets.
- The paper is well-organized and clearly presented.
- In addition to empirical evidence, the paper provides theoretical proofs to support the advantages of the proposed method.
- The provided supplementary material contains code, dataset and checkpoints. This suggests good reproducibility of the results in the paper.

**Weaknesses:**

- As I mentioned in the summary, the problem is meaningful yet less tackled in the AI community. To my eyes, it is mostly due to the missing prerequisites of biological knowledge. The paper provides some explanation of the problem in the introduction, but it would be much more helpful if more context could be provided (maybe in the appendix).
- Line 215: The motivation for computing edge weights on random walk paths is unclear, and no empirical evidence is provided to support it. Since the context graph incorporates data from three modalities, the edges likely exhibit strong heterogeneity. Is there evidence suggesting that a cumulative product of edge weights effectively captures dependency or similarity between nodes?
- Line 303: The approach for avoiding noisy edges in computations lacks motivation and an ablation study. Providing more detail here would offer valuable insights for researchers interested in extending InfoAlign to new contexts.

**Questions:**

- What are the minimum data requirements for cell morphology and gene expression to effectively apply InfoAlign? How does the method perform when these data are sparse or incomplete?
- Given the promising performance of structure-based pretrained GNNs, have the authors considered using representations from these pretrained models instead of Morgan fingerprints as molecular features? This might leverage the strengths of both approaches.
- How does the computational complexity of InfoAlign compare to existing methods, and what are the practical limitations in terms of dataset size and computational resources?

---

> ### Author Response · Authors · 2024-11-18
> **Author Response [1/2]**
>
> We sincerely appreciate the reviewer's thoughtful suggestions and questions. We have provided point-by-point answers to each weakness and question. We have also revised the main text and appendix to incorporate the reviewer's valuable feedback, with all changes clearly highlighted in blue for ease of reference. Should any concerns remain, we remain fully committed to addressing them promptly and thoroughly.
>
> ## W1: Biological context
>
> Thank you for recognizing the significance of our research problem. In the revision, we have further clarified the definition of the tasks (Appendix D.1) to reduce the barrier to understanding the biological knowledge prerequisites.
>
> So far, we have: (1) formulated the problem as a machine learning task in Section 2, (2) explained the method and curated the dataset for ease of use in Sections 5.1, 6, and Appendices C and D, (3) provided a clear background on the motivation in the first paragraph of the Introduction, and (4) expanded the discussion of cellular response data in the related work section, Appendices C and D, with relevant references for readers interested in further details.
>
> We hope these efforts will help readers in the ML community better understand the problem and encourage further solutions. We would also be happy to discuss additional strategies to further clarify the biological prerequisites in the main text and appendix.
>
> ## W2: Edge weight
>
> ### Edge weights are motivated by tasks in drug discovery.
>
> Sorry for the confusion: We have now clarified this in the revision (Lines 198-201):
> "For example, edges derived from computational criteria between molecule nodes are assigned weights based on the assumption that structurally similar molecules may exhibit similar biological effects, a concept widely used in drug discovery, such as lead optimization."
>
> Specifically, as introduced in Section 5, edges based on chemical and biological criteria have uniform weights (i.e., edge weights are set to 1). Edge weights are primarily applied to edges introduced by computational criteria, where we compute similarity using features, ranging from 0 to 1. For example, the similarity between Morgan fingerprints constructs weighted edges in the context graph. This approach is motivated by assumptions in tasks like lead optimization, where structurally similar molecules may exhibit similar biological effects. The weights quantify similarity, rather than assuming identical effectiveness for structurally similar molecules.
>
> ### Edge weights are observed to empirically improve performance robustness.
>
> In pretraining, the cumulative edge weights help avoid aligning a molecule with features from distant nodes on the context graph, especially when using longer walk lengths. In the table below, we present new experiments on ChEMBL2K using an unweighted context graph (all edge weights set to 1). With walk length $L=4$ and weighted edges, InfoAlign achieves 81.3±0.6. As the walk length increases, the performance with weighted edges is more stable and performs better than with unweighted edges.
>
> | Walk Length       | L=8           | L=10          | L=12          |
> |--------------|---------------|---------------|---------------|
> | weighted     | 80.28±0.58    | 81.14±0.32    | 81.15±0.56    |
> | unweighted   | 79.75±0.57    | 79.57±0.51    | 79.94±0.36    |
>
> ## W3: Remove noisy edges
>
> Thanks for your suggestion. We conducted new experiments without using the mechanism described in Section 5 to remove noisy edges. As shown in the first row of the table below, performance decreases to varying degrees with more noisy edges, compared to using the mechanism with fewer noisy edges.
>
> |                  | ChEMBL2K     | Broad6K      | ToxCast      | Biogen3K     |
> |-------------------------|--------------|--------------|--------------|--------------|
> | with more noisy edges     | 79.97±0.21   | 69.03±0.22   | 65.88±0.92   | 50.97±0.62   |
> | with fewer noisy edges     | 81.33±0.62   | 69.95±0.09   | 66.36±1.05   | 49.42±0.18   |
>
> (The setting for "with more noisy edges": after computing the similarity, we applied a threshold of 0.5 to select the similar edges.)

---

> ### Author Response · Authors · 2024-11-18
> **Author Response [2/2]**
>
> ## Q1: Data requirements
>
> ### InfoAlign does not require cell morphology and gene expression data for downstream tasks but improves InfoAlign pretraining
>
> Cellular response data support the pretraining of InfoAlign. For downstream tasks, where only molecules are used as input and the output is a molecular representation, no additional data types are required.
>
> We have updated Appendix D.4 with new experiments that removed either cell morphology or gene expression-related nodes from the context graph. The results are also given in the table below. The downstream performance drops when these nodes are removed. Although gene expression data is more sparse (about ten times less than cell morphology), it still contributes valuable information to InfoAlign’s performance. In summary, we believe that additional data from cell morphology and gene expression add value, and InfoAlign may perform worse with less data. Fortunately, we have curated multiple sources of cellular response data to ensure diverse and comprehensive data for pre-training.
>
>
> |                      | ChEMBL2K     | Broad6K      | ToxCast      | Biogen3K     |
> |-----------------------------|--------------|--------------|--------------|--------------|
> | w/o cell-related nodes      | 79.57±0.58   | 68.41±0.31   | 65.11±0.82   | 51.21±0.17   |
> | w/o gene-related nodes      | 77.97±0.33   | 67.1±0.17    | 64.93±0.96   | 51.57±0.46   |
> | InfoAlign                   | 81.33±0.62   | 69.95±0.09   | 66.36±1.05   | 49.42±0.18   |
>
>
> ## Q2: Molecular features on context graphs
>
> Thank you for the interesting question. In this work, we primarily use Morgan fingerprints to avoid introducing unnecessary factors that could influence the development and analysis of InfoAlign's pre-training strategy and model. Based on our observations (Table 1/2), Morgan fingerprints remain a competitive and cost-effective representation of molecular structures.
>
> Improving InfoAlign with other pre-trained GNN representations or even their ensemble is indeed a promising direction. However, since this is outside the main focus of this work, we leave it for future exploration.
>
> ## Q3: Computational Complexity
>
> Thank you for the question. Compared to existing work, InfoAlign has cost dealing with the context graph. Pretraining with the context graph introduces minimal additional computational complexity. We use a sparse matrix to extract the random walk on the context graph. Let $N$ and $M$ denote the number of nodes and edges in the context graph, respectively, with an average node degree $k$, where $k \ll N$ and $M \ll N$. The time and space complexity of the random walk are $\mathcal{O}(k)$ and $\mathcal{O}(M)$, both much smaller than the dense version's $\mathcal{O}(N^2)$.
>
> By using random walks to sample local neighborhoods for pretraining molecules, we can scale the context graph efficiently. Data size is not a major concern; the practical limitation lies in the limited number of cell morphology and gene expression features due to the high cost of generating them. Pretraining is efficient on a V100, using less than 13GB of GPU memory with a large batch size of 3072.

---

> > ### Comment · Reviewer_hxs4 · 2024-11-24
> >
> > Thank you for the clarifications. I would like to keep the current scores.

---

> > > ### Author Response · Authors · 2024-11-25
> > > **Thank you for your response**
> > >
> > > Thank you for your response. We sincerely appreciate the reviewers' insightful comments provided in the rebuttal and believe we have addressed them point-by-point. If there are any remaining concerns that the reviewers feel have not been fully addressed, we would be grateful for the opportunity to discuss them further during the discussion phase.

---

### Meta-Review · Area_Chair_fdaj · 2024-12-21

**Metareview:**

This paper proposes a new approach to learn molecular representations through incorporating the information of cellular responses. This is done by (1) constructing a context graph for cellular response data and (2) information bottleneck training on the extracted random walk.

Overall, I find the paper to provide meaningful and solid contribution to the field via introducing a (relatively) new data modality. The experiments are convincing (despite some room for improvement). This paper is likely to promote some future works to consider learning molecular representation using cellular responses.

There was a valid concern raised by the reviewers, that the main experiments are conducted without fine-tuning the model. I tend to agree with the concern and it would have been better for the authors to fully explore the performance of models after finetuning. Especially, the authors could have selected a subset of the considered baselines and compared with them after finetuning. However, I believe this concern is partially alleviated by the new experiments constructed by the authors during the rebuttal. I also believe that the old experiments still convincingly show the promise of the new idea.

Overall, I recommend acceptance for this paper since it introduces a meaningful data modality to molecular representation learning with meaningful results. The highly encourage the authors to extend their evaluation to fine-tuning setting which better aligns with the practical scenarios.

**Additional Comments On Reviewer Discussion:**

Reviewer Sxqu raised strong concerns on the empirical evaluation of the paper. I agree with these points and the experiments are not sufficient to verify if the model achieves SOTA (the authors do not claim SOTA). However, I do not think all the methods need to achieve SOTA to be published, especially for a crowded research area like molecular representation learning. I also think SOTA in this field is less meaningful since public data is limited and it is hard to scale the models.

---

### Decision · Program_Chairs · 2025-01-22

Accept (Poster)